



# The potential role of methanesulfonic acid (MSA) in aerosol formation and growth and the associated radiative forcings

Anna L. Hodshire[1], Pedro Campuzano-Jost[2,3], John K. Kodros[1], Betty Croft[4], Benjamin A. Nault[2,3], Jason C. Schroder[2,3], Jose L. Jimenez[2,3], Jeffrey R. Pierce[1]

[1]Department of Atmospheric Science, Colorado State University, Fort Collins, CO 80523, USA
[2]Department of Chemistry, University of Colorado, Boulder, CO, USA
[3]Cooperative Institute for Research in Environmental Sciences, University of Colorado, Boulder, CO, USA
[4]Dalhousie University, Department of Physics and Atmospheric Science, Halifax, NS, B3H 4R2, Canada

*Correspondence to*: Anna L. Hodshire (Anna.Hodshire@colostate.edu)

**Abstract.** Atmospheric marine aerosol particles impact Earth's albedo and climate. These particles can be primary or secondary and come from a variety of sources, including sea salt, dissolved organic matter, volatile organic compounds, and sulfur-containing compounds. Dimethylsulfide (DMS) marine emissions contribute greatly to the global biogenic sulfur budget, and its oxidation products can contribute to aerosol mass, specifically as sulfuric acid and methanesulfonic acid (MSA). Further, sulfuric acid is a known nucleating compound, and MSA may be able to participate in nucleation when bases are available. As DMS emissions, and thus MSA and sulfuric acid from DMS oxidation, may have changed since pre-industrial times and may change in a warming climate, it is important to characterize and constrain the climate impacts of both species. Currently, global models that simulate aerosol size distributions include contributions of sulfate and sulfuric acid from DMS oxidation, but to our knowledge, global models typically neglect the impact of MSA on size distributions.

In this study, we use the GEOS-Chem-TOMAS (GC-TOMAS) global aerosol microphysics model to determine the impact on aerosol size distributions and subsequent aerosol radiative effects from including MSA in the size-resolved portion of the model. The effective equilibrium vapor pressure of MSA is currently uncertain, and we use the Extended Aerosol Inorganics Model (E-AIM) to build a parameterization for GC-TOMAS of MSA's effective volatility as a function of temperature, relative humidity, and available gas-phase bases, allowing MSA to condense as an ideally nonvolatile or semivolatile species or too volatile to condense. We also present two limiting cases for MSA's volatility, assuming that MSA is always ideally nonvolatile (irreversible condensation) or that MSA is always ideally semivolatile (quasi-equilibrium condensation but still irreversible condensation). We further present simulations in which MSA participates in binary and ternary nucleation with the same efficacy as sulfuric acid whenever MSA is treated as ideally nonvolatile. When using the volatility parameterization described above (both with and without nucleation), including MSA in the model changes the global annual averages at 900 hPa of submicron aerosol mass by 1.2%, N3 (number concentration of particles greater than 3 nm in diameter) by -3.9% (non-nucleating) or 112.5% (nucleating), N80 by 0.8% (non-nucleating) or 2.1% (nucleating), the aerosol indirect effect (AIE) by -8.6 mW m$^{-2}$ (non-nucleating) or -26 mW m$^{-2}$ (nucleating), and the direct radiative effect (DRE) by -15 mW m$^{-2}$ (non-nucleating) or -14 mW m$^{-2}$ (nucleating). The sulfate and sulfuric acid from DMS oxidation produces 4-6 times more submicron mass than MSA does, leading to ~10 times a stronger cooling effect in the DRE. But the



changes in N80 are comparable between the contributions from MSA and from DMS-derived sulfate/sulfuric acid, leading to comparable changes in the AIE.

Model-measurement comparisons with the Heintzenberg et al. (2000) dataset over the Southern Ocean indicate that the default model has a missing source or sources of ultrafine particles: the cases in which MSA participates in nucleation
(thus increasing ultrafine number) most closely match the Heintzenberg distributions, but we cannot conclude nucleation from MSA is the correct reason for improvement. Model-measurement comparisons with particle-phase MSA observed with a customized Aerodyne high-resolution time-of-flight aerosol mass spectrometer (AMS) from the ATom campaign show that cases with the MSA volatility parameterizations (both with and without nucleation) tend to fit the measurements the best (as this is the first use of MSA measurements from ATom, we provide a detailed description of these measurements and their
calibration). However, no one model sensitivity case shows the best model-measurement agreement for both Heintzenberg and the ATom campaigns. As there are uncertainties in both MSA's behavior (nucleation and condensation) and the DMS emissions inventory, further studies on both fronts are needed to better constrain MSA's past, current and future impacts upon the global aerosol size distribution and radiative forcing.

# 1 Introduction

Atmospheric marine particles contribute significantly to the global aerosol budget and impact the planetary albedo and climate (Quinn et al., 2015; Reddington et al., 2017). The number concentration, size, and chemical composition of these marine particles determine their ability to affect climate, through either absorbing and scattering incoming solar radiation
(the direct radiative effect [DRE]; Charlson et al., 1992; Erlick et al., 2001) or indirectly, by modifying cloud properties (the cloud-albedo aerosol indirect effect [AIE]; de Leeuw et al., 2011). For the DRE, the magnitude and relative division between absorbing and scattering will depend on both the particle size and composition (Bond et al., 2006; 2013); peak efficiencies for scattering and absorbing solar radiation are typically reached with particles between 100 nm to 1 μm in diameter (Seinfeld and Pandis, 2006). The AIE refers to aerosols' ability to alter cloud properties, including the reflectivity (albedo) of
clouds by changing properties such as the cloud droplet number concentration (CDNC) (Twomey, 1974). Typically, particles act as cloud condensation nuclei (CCN) if they are larger than 40-100 nm; the ability of a particle to act as a CCN is also dependent upon particle hygroscopicity (Petters and Kreidenweis, 2007). The number of particles in these size ranges depend on primary emissions, as well as nucleation, condensation, and coagulation (Pierce and Adams, 2009a). To improve model estimates of the DRE and AIE, models must account for nucleation and condensational growth from marine particles, as for
a significant portion of the remote atmosphere, marine and stratospheric particles are the main sources of particles (e.g. Huang et al., 2013).





Biologically productive oceans emit organic compounds such as VOCs, primary biological particles, primary organic particles, and halocarbons (Quinn et al., 2015). Sources of marine particles often indicate organic species present (e.g. Heintzenberg et al., 2001; O'Dowd et al., 2007; Frossard et al., 2014, Wang et al., 2017) that could dominate submicron aerosol mass (O'Dowd et al., 2004; Facchini et al., 2008). Sulfur-containing organic compounds in the form of

dimethylsulfide (DMS; $CH_3SCH_3$) and organosulfates (Bates et al., 1992, Quinn et al., 2015) are an important source of marine emissions. DMS emissions constitutes approximately 50% of the global biogenic sulfur budget as currently understood (e.g. Andreae, 1990) and approximately 21% of all sulfur precursor gases (Textor et al., 2006). DMS and its oxidation products have been the focus of many studies determining the gas-phase chemistry (e.g. Barnes et al. 2006 and references therein), gas-phase kinetics (e.g. Wilson and Hirst, 1996 and references therein), and possible impact to the

aerosol size distribution and radiative budget (e.g. Korhonen et al., 2008; Woodhouse et al., 2013). Much of this research has stemmed from efforts to test the hypothesis that DMS emissions may regulate climate through a temperature-emissions feedback (the CLAW hypothesis, Charlson et al. (1987)).

The main products of DMS from oxidation by the hydroxyl radical are sulfur dioxide ($SO_2$) and methanesulfonic acid ($CH_3S(O)_2OH$, MSA) (Andreae et al., 1985). $SO_2$ can further oxidize to create sulfuric acid ($H_2SO_4$). The effective

equilibrium vapor pressure of sulfuric acid in the presence of water in the troposphere is negligible compared to sulfuric acid concentrations under all atmospherically relevant conditions (Marti et al., 1997), allowing sulfuric acid to readily condense onto particles of all sizes and participate in particle nucleation (e.g. Kulmala et al., 2000). Gas-phase concentrations of MSA have been observed to be 10-100% of sulfuric acid concentrations in coastal marine boundary layers (Eisele and Tanner, 1993; Berresheim et al., 2002; Mauldin et al., 2003), and MSA can contribute to the growth of pre-existing marine particles,

at times contributing over half as much bulk aerosol mass as non-sea salt sulfate to the total aerosol burden (e.g. Preunkert et al., 2008; Legrand et al., 2017). To our knowledge, the effective equilibrium vapor pressure of MSA, which should depend on temperature, relative humidity, and availability of bases, has not previously been well-quantified for the range of potential atmospheric conditions. Also as yet, MSA has not yet been observed in the field to directly contribute to aerosol nucleation, although Dall'Osto et al. (2018) observed new particle formation events over Greenland that suggest that MSA could be

involved in a portion of the events. Bork et al. (2014) determined through the Atmospheric Cluster Dynamics Code kinetic model (McGrath et al., 2012; Olenius et al., 2013) that the presence of MSA could increase the molecular cluster formation rates by as much as one order of magnitude for a $MSA$-$H_2SO_4$-DMA (DMA = dimethylamine) system under atmospherically relevant MSA concentrations. This enhancement is predicted to be typically less than 300% at 258 K and less than 15% at 298 K for the case of [DMA] = $10^9$ molecules cm$^{-3}$ (Bork et al., 2014). Chen et al. (2015) observed an $MSA$-$H_2O$-TMA

(TMA = trimethylamine) system to nucleate in the laboratory, but at an efficiency lower than that of the $H_2SO_4$-$H_2O$ system; Chen and Finlayson-Pitts (2017) further observed nucleation of $MSA$/$H_2O$ systems with TMA, DMA, MA (MA = methylamine) and ammonia. To our knowledge, global models that simulate aerosol number concentrations (e.g. D'Andrea et al. 2013; Kodros et al., 2018; Ma and Yu, 2015; Regayre et al., 2018; Xausa et al., 2018) only track the effect of sulfuric acid and aqueous sulfate from DMS/$SO_2$ oxidation on the aerosol size distribution and not MSA. Thus, the potential




contribution towards nucleation and/or size-resolved particle growth by MSA and the resulting radiative impacts has not yet been quantified.

The effective volatility (equilibrium vapor pressure above the particle-phase mixture) of MSA will modulate its impact on the aerosol size distribution. Condensational growth of vapors to the particle phase is controlled by both the

volatility of the condensing species and the concentration of the species in the gas phase. Riipinen et al. (2011) presented two limiting cases of growth for gas-phase condensable material:

(1) Compounds with low enough saturation vapor concentrations (C*; Donahue et al., 2006) may be considered essentially nonvolatile to condense irreversibly through kinetic, gas-phase-diffusion-limited condensation (Riipinen et al. 2011; Zhang et al., 2012). This type of growth is referred to as "kinetic condensation" by Riipinen et al. (2011); it can be

thought of as effectively nonvolatile condensation. The effective volatility required to achieve effectively nonvolatile condensation typically must be less than $C* \lll 10^{-3}$ $\mu$g m$^{-3}$ (e.g. low and extremely low volatility organic compounds; LVOCs and ELVOCs) (Pierce et al., 2011; Donahue et al., 2011) and the contribution to growth from effectively nonvolatile condensation is proportional to the Fuchs-corrected particle surface area (Pandis et al., 1991). We will refer to this type of condensation as "ELVOC-like" condensation in this work.

(2) In contrast, semi-volatile species (e.g. semi-volatile organic compounds; SVOCs) with average C* between $10^0$-$10^2$ $\mu$g m$^{-3}$ (Murphy et al., 2014) quickly reach equilibrium between gas and particle phases for all particle sizes. As a result, the contribution to growth is proportional to the aerosol mass distribution (Pierce et al., 2011; Riipinen et al., 2011; Donahue et al., 2011; Zhang et al., 2012), limiting the growth of ultrafine particles. This type of growth is referred to as "thermodynamic condensation" by Riipinen et al. (2011) and "quasi-equilibrium" growth by Zhang et al. (2012); we will

refer to this type of condensation as "SVOC-like" condensation in this work.

Important characteristics for growth in these regimes is that under ELVOC-like condensation, particles in the kinetic regime ($D_p$ < ~50 nm) all grow in diameter at the same rate (e.g. nm h$^{-1}$) regardless of diameter, whereas in the continuum regime (Dp > ~1 $\mu$m), particle growth rates are proportional to $1/D_p$. Conversely, SVOC-like condensation growth rates scale with $D_p$ for all particle sizes, favoring the largest particles. Thus, if MSA participates in ELVOC-like

condensation, ultrafine particles are able to grow more quickly to climatically relevant sizes (e.g. CCN) as compared to SVOC-like condensation. In reality, MSA's contribution towards growth likely lies between these two limiting cases: as MSA is an acid, its volatility will depend on not only temperature but also relative humidity and gas-phase bases (e.g. Barsanti et al., 2009; Yli-Juuti et al., 2013; Hodshire et al., 2016).

In this study, we use the GEOS-Chem-TOMAS global chemical transport model to estimate the contribution of

MSA to the aerosol size distribution and resulting radiative effects. We examine (1) MSA condensation assumptions, testing the limiting cases of growth (ELVOC-like vs SVOC-like) as well as a parameterization of volatility dependent on temperature, water vapor, and gas-phase bases built from a phase-equilibrium model and (2) how the contribution of MSA changes depending on whether or not it is allowed participate in nucleation. We further use global measurements of aerosol size distributions as compiled by Heintzenberg et al. (2000) and MSA mass as observed on the ATom mission to compare




the various model assumptions. Our goals are to determine the sensitivity of the aerosol size distribution and radiative impacts implied by the various assumptions, and to see if the assumptions can be constrained by observations. This study is a first look at how MSA might impact the global aerosol size distribution and associated climate effects by considering the sensitivity of its assumed volatility and ability to impact nucleation. Along with our model analyses of MSA, we provide a

detailed overview of the calibration applied to an Aerodyne high-resolution time-of-flight aerosol mass spectrometer (AMS) for detecting MSA during the ATom mission in the supplemental information as a general reference for the AMS community.

## 2 Methods

### 2.1 Model description

In this work, we use the GEOS-Chem chemical transport model version 10.01 (http://geos-chem.org) coupled to the online TwO-Moment Aerosol Sectional (TOMAS) microphysical module (Adams and Seinfeld, 2002; GEOS-Chem-TOMAS as described in Kodros et al., 2016; 2017) to test the sensitivity of the aerosol size distribution to the addition of a

marine secondary organic aerosol (SOA), represented in this work by methanesulfonic acid (MSA), of varying effective volatility and nucleation capability. The version of GEOS-Chem-TOMAS (GC-TOMAS) used here has 47 vertical levels, a horizontal resolution of 4°x5° (~400 km at mid latitudes), and GEOS-FP reanalysis (http://gmao.gsfc.nasa.gov) for meteorological inputs. GC-TOMAS uses 15 size sections spanning dry diameters from approximately 3 nm - 10 μm and explicitly tracks total particle number as well as sulfate, sea salt, dust, hydrophilic OA, hydrophobic OA, internally mixed

BC, externally mixed BC, and water mass (Lee and Adams, 2012). Biomass burning emissions are simulated using the Fire INventory from NCAR version 1.0 (FINNv1) (Wiedinmyer et al., 2011). Dust emissions follow the parameterization of the DEAD scheme (Zender et al., 2003); sea-salt aerosol emissions follow the parameterization of Jaegle et al. (2011). Anthropogenic emissions are from the Emissions Database for Global Atmospheric Research (EDGAR; Janssens-Maenhout et al., 2010). Regional EDGAR overwrites are used as follows: black and organic carbon emissions from fossil-fuel and

biofuel combustion processes are from Bond et al. (2007). Grid-box gas-phase concentrations of $NH_3$ are used in determining the volatility regime of MSA in the MSA parameterization (Sect. 2.2): global anthropogenic, biofuel, and natural ammonia sources are from the Global Emissions InitiAtive (GEIA) (Bouwman et al., 1997). In Europe, Canada, the U.S., and Asia, anthropogenic emissions are overwritten by the European Monitoring and Evaluation Programme (Centre on Emissions Inventories and Projections, 2013), the Criteria Air Contaminant Inventory (http://www.ec.gc.ca/air/default.asp?

lang=En&n=7C43740B-1), the National Emission Inventory from the U.S. EPA ((http://www.epa.gov/ttnchie1/net/2011inventory.html), and the MIX (Li et al., 2017) inventories, respectively. All simulations are run for 2014, with one month of model spinup that is not included in the analysis. All results are presented as annual or monthly averages.




We use the default (at the time of this model version) GEOS-Chem DMS emissions inventory (Kettle et al., 1999; Kettle and Andreae 2000) for this study. We acknowledge that the updated DMS inventory of Lana et al. (2011) includes more up-to-date measurements than the default DMS inventory for GEOS-Chem v10.01. Their work found that the default climatology overpredicted DMS emissions in some latitudes/seasons but underpredicted DMS emissions in other

latitudes/seasons. We found, however, that using the Lana emission inventory led to minor differences in MSA impacts spatially but overall, similar magnitudes of changes were observed. See the supplement Sect. S2 for more analysis of the two different emissions inventories.

In the standard GEOS-Chem DMS mechanism, DMS reacts with OH through the OH addition pathway to form molar yields of 0.75 $SO_2$ and 0.25 MSA (Chatfield and Crutzen, 1990; Chin et al., 1996). DMS also reacts with the nitrate

radical ($NO_3$) to form a molar yield of 1 $SO_2$. $SO_2$ can then (1) react further in the model with OH to form gas-phase sulfuric acid, (2) undergo aqueous oxidation with $H_2O_2$ or $O_3$ to form condensed sulfate, or (3) be lost through dry and wet deposition processes (Pierce et al., 2013). Pierce et al. (2013) found that in GC-TOMAS (v8.02.02), 26% of global SO2 formed sulfate through aqueous chemistry and 13% formed sulfuric acid through gas-phase reaction with OH (the rest was lost through dry and wet deposition). The sulfate formed through aqueous chemistry is added to CCN-sized particles when activated in

clouds, whereas the sulfuric acid formed from OH reactions participates in nucleation and irreversible condensation to particles of all sizes. Prior to this work, the DMS/$SO_2$-oxidized sulfuric acid and sulfate was included in the size-resolved portion of the GC-TOMAS model but MSA was not. In this study, we include MSA in the size-resolved microphysics of the model. The contribution of MSA from DMS towards the sulfate budget and the size distribution as a function of particle size will then depend on both MSA's volatility and ability to participate in nucleation, as discussed below. A discussion of

alternative oxidation pathways of DMS and the potential importance of aqueous-phase DMS chemistry (currently not included in GEOS-Chem) is provided in Sect 2.6.

Nucleation is simulated via a ternary nucleation scheme involving water, sulfuric acid, and ammonia (Napari et al., 2002), scaled with a global tuning factor of $10^{-5}$ (Jung et al., 2010; Westervelt et al., 2013). In ammonia-limited regions (less than 1 pptv), a binary nucleation scheme involving water and sulfuric acid (Vehkamaki et al., 2002) is instead used. When

MSA is assumed to participate in nucleation, it is treated as an extra source of sulfuric acid for the ternary nucleation scheme within the model. Growth and loss of nucleated particles between 1 and 3 nm is simulated using the parameterization of Kerminen et al. (2004) (Lee et al. 2013) with growth in this size range controlled by the pseudo-steady-state sulfuric acid (Pierce and Adams, 2009b) and MSA when it participates in nucleation.

SOA in GC-TOMAS is traditionally formed from terrestrial biogenic sources, with the biogenic source represented

by 10% of the monoterpene emissions, totalling to 19 Tg(SOA) yr$^{-1}$; we further include 100 Tg(SOA) yr$^{-1}$ spatially correlated with CO to represent anthropogenic SOA and anthropogenically-controlled biogenic SOA (Spracklen et al., 2011; D'Andrea et al., 2013). The default GC-TOMAS setting is for SOA to form through effective nonvolatile condensation (ELVOC-like condensation) onto pre-existing particles at the time of emission of the parent compound. However, it is possible to instead have SOA form in GC-TOMAS through quasi-equilibrium condensation (SVOC-like condensation, but



still irreversible, e.g. not allowing for re-evaporation, in the model) by distributing the SOA across aerosol sizes proportional to the aerosol mass distribution. In this work, we assuming ELVOC-like SOA condensation as it performed best relative to size-distribution measurements in D'Andrea et al. (2013).

**2.2 MSA volatility assumptions, calculations, and parameterization**

As the effective volatility of MSA is uncertain, we use the Extended Aerosol Inorganics Model (E-AIM (http://www.aim.env.uea.ac.uk/aim/aim.php, Clegg et al., 1992; Clegg and Seinfeld, 2006a, b; Wexler and Clegg, 2002)) to build a parameterization for GC-TOMAS of MSA's potential volatility as a function of temperature, relative humidity, and available gas-phase bases. E-AIM calculates the MSA equilibrium vapor pressure above the particle mixture ($C_{eq}$ in units of

$\mu g\ m^{-3}$), and thus we get an MSA volatility parameterization in terms of $C_{eq}$ (Fig. 1). We also consider two ideal assumptions of MSA volatility: (1) MSA condenses as and ELVOC-like species, condensing irreversibly to aerosol of all sizes, with net condensation of MSA proportional to the Fuchs-corrected aerosol surface area. Conversely, (2) MSA condenses as an SVOC-like species, where the net condensation of MSA is proportional to the aerosol mass distribution.

In order to use a potentially more realistic representation in the model of a possible range of volatilities for MSA, a

parameterization for the volatility of MSA was derived using the Extended Aerosol Inorganics Model (E-AIM). As MSA is a strong acid (pKa=-1.96; Haynes, 2017), we must consider the amount of atmospheric gas-phase base present; ammonia is used in E-AIM as the representative base. Although Chen and Finlayson-Pitts (2017) found in laboratory experiments that MSA had different rates of new particle formation with amines than ammonia, GC-TOMAS currently does not include any amine concentrations and thus we do not attempt to account for these variations. Figure S1 and S2 provides global annual

and seasonally averaged $NH_3$ concentrations from GEOS-Chem-TOMAS. The effective volatility of MSA also depends on the ambient temperature (Donahue et al. 2006) and relative humidity (RH) (Chen et al., 2018). We run E-AIM for between 10-100% RH and between 240-310 K. Figure 1 shows the resulting volatility as a function of RH and temperature for conditions with no free ammonia and excess ammonia (3 times as many moles of free ammonia than moles of MSA). At low-base conditions (Fig. 1a), MSA acts essentially as a VOC (will all stay in vapor phase) below 90% RH and condenses as

an ideally SVOC-like species above 90% RH for the entire input temperature range. Conversely, for excess-base conditions, we see that MSA transitions between volatilities as a function of both temperature and RH. We parameterize a transition between ELVOC-like behavior and SVOC-like behavior for excess-base conditions along the $C_{eq} = 10^{-2}\ \mu g\ m^{-3}$ line using the dashed line in Fig. 1b, given by:

$$T_{trans}(\text{RH}) = a - b \cdot RH + c \cdot RH^2 - d \cdot RH^3 + e \cdot RH^4 \,, \tag{1}$$

where *RH* is the relative humidity, *T* is the temperature, $T_{trans}$ is the transition temperature, and *a, b, c, d,* and *e* are fit coefficients, whose values are listed in Table 1. If $T > T_{trans}$, then MSA is treated as an ideally SVOC-like species that undergoes quasi-equilibrium condensation in GC-TOMAS. If $T < T_{trans}$, then MSA is low to extremely low in volatility and



will be treated as an ideally ELVOC-like speces that undergoes gas-phase-diffusion-limited condensation in GC-TOMAS. We do not include a volatile region under excess-base conditions: the high-temperature, low-RH regions that this would be applicable to are globally limited and likely only occur over desert regions, where MSA formation is likely negligible. Although E-AIM predicts that MSA's volatility varies smoothly across the volatility space as a function of

temperature and RH, for simplicity, we only assume three condensational regimes: SVOC-like condensation, ELVOC-like condensation, and VOC-like (no condensation).

When using this parameterization in GC-TOMAS, we use a gas-phase ammonia mixing ratio of 10 pptv as a cutoff between the no ammonia and excess ammonia cases as this roughly marks the transition from acidic to neutral aerosol (Croft et al., 2016, Supplementary Fig. 4). The gas-phase MSA production rate is explicitly tracked in the model, but not the MSA

gas-phase concentrations. At the time of production, the model will then determine whether to treat MSA condensation as an effectively volatile species (no MSA condensing), an SVOC-like species (with all of the MSA produced condensed to the mass distribution), or an ELVOC-like species (with all of the MSA produced condensing the the Fuchs surface area and participating in the nucleation calculation in some simulations), based on the current $T$, RH, and available ammonia. For both SVOC-like and ELVOC-like condensation, the condensation is irreversible; we do not let MSA partition back to the gas

phase once it is condensed as we do not track gas-phase MSA in the model. Even this simple parameterization is a significant increase in the physical representation of MSA volatility over assuming a fixed volatility.

### 2.3 Descriptions of simulations

The different GEOS-Chem-TOMAS (GC-TOMAS) simulations of this study are summarized in Table 2. The

default (DEFAULT_NoMSA) simulation represents a default GEOS-Chem-TOMAS simulation with only sulfate and sulfuric acid from DMS/SO$_2$ oxidation included in TOMAS; DEFAULT_NoMSA will be the comparison simulation for all other cases. PARAM_NoNuc uses the volatility parameterization from E-AIM (Sect 2.2), treating MSA as a non-nucleating ELVOC, an SVOC, or a VOC, depending upon the temperature, RH, and amount of ammonia in the gas-phase. ELVOC_NoNuc treats MSA condensation as ELVOC-like condensation. SVOC_NoNuc treats MSA condensation as

SVOC-like (but irreversible, Section 2.2). PARAM_Nuc and ELVOC_Nuc are identical to PARAM_NoNuc and ELVOC_NoNuc except that MSA is allowed to participate in nucleation with the properties of sulfuric acid, providing an upper bound on the role of MSA in nucleation. For PARAM_Nuc, MSA only participates in nucleation when MSA is in the ELVOC-like regime; for ELVOC_Nuc, MSA is always able to participate in nucleation. Finally, to determine the contribution of sulfate and sulfuric acid from DMS/SO$_2$ oxidation alone to the default size distribution, we run a case with

DMS emissions turned off (NoDMS_NoMSA).

In the Supplementary Information, we test the sensitivity of the model to the DMS concentration with two additional DMS inventories: the first is the DMS emissions inventory of Lana et al. (2011) and the second is the default DMS emissions inventory increased globally by a factor of two. As the sulfate and sulfuric acid from DMS/SO$_2$ oxidation is included in the default case simulation, we run new default simulations with the new DMS inventories





(DEFAULT_NoMSA_Lana and DEFAULT_NoMSA_2xDMS). We use the PARAM_NoNuc case settings to determine the change in MSA's impact to the size distribution under the new DMS emissions inventories (PARAM_NoNuc_Lana and PARAM_NoNuc_2xDMS). However, the results for the contribution of MSA to the size distribution do not qualitatively change between the default DMS emissions inventory and the Lana DMS emission inventory. The contribution of MSA

towards the submicron aerosol mass and thus the aerosol DRE in the 2xDMS case is roughly double that of the 1xDMS case but N3 and N80 do not significantly change for our tested metrics. Hence, we will not include these model results in the main portion of the paper. See the supplementary information, Sect. S2, Tables S1-S2, and Figs. S3-S5 for a brief analysis of the different inventories.

**2.4 Analysis of simulated radiative effects**

We calculate aerosol DRE and AIE following Kodros et al., (2016). The all-sky DRE is calculated offline using the monthly mean aerosol mass and number distributions from the GC-TOMAS output. The refractive indices are from GADS (Global Aerosol Dataset; Koepke et al., 1997). Aerosol optical depth (AOD), single-scattering albedo, and the asymmetry parameter are calculated from Mie code (Bohren and Huffman, 1983). Optical properties and the monthly mean albedo and

cloud fractions from GEOS5 are used as inputs to the offline version of the Rapid Radiative Transfer Model for Global Climate Models (RRTMG: Iacono et al., 2008) that has been implemented for the standard (non-TOMAS) version of GEOS-Chem (Heald et al., 2014). We assume an internal mixture, spherical particles, non-absorptive OA (brown carbon is not considered in this work) and a core-shell morphology. We note that the mixing state may vary both regionally and temporally, and that using only one mixing state globally for the full year is a limitation of our analysis of the DRE.

The AIE is calculated as follows: first, the CDNC is found using the activation parameterization of Abdul-Razzak and Ghan (2002) for the monthly mean aerosol mass and number distribution from the GC-TOMAS output. A constant updraft velocity of 0.5 m s$^{-1}$ is assumed. We again assume the aerosol species are internally mixed within each TOMAS size bin to determine $\kappa$, the hygroscopicity parameter, as a volume-weighted average of the individual aerosol species (Petters and Kreidenweis, 2007). For the AIE, we use an effective cloud drop radii of 10 µm as a control and then perturb this value

with the ratio of the CDNC of each sensitivity case to the default case to the one-third power, following the methods of Rap et al., (2013), Scott et al., (2014), and Kodros et al., (2016):

$$r_{perturbed} = \left(\frac{CDNC_{base\ case}}{CDNC_{sensitivity\ case}}\right)^{1/3} \cdot 10\mu m \ , \qquad (2)$$

RRTMG is again used to determine the changes in the top-of-the-atmosphere radiative flux from the changes in effective cloud drop radii, with monthly mean meteorological data needed as inputs again informed by GEOS5. For more details on

the methods used for the DRE and AIE calculations, refer to Kodros et al. (2016) and references therein.



### 2.5 Measurement comparisons

Heintzenberg et al. (2000) compiled 30 years (between ~1970-1999) of physical marine aerosol data from both sampling sites and field campaigns to create annual global size distribution parameters, fitting the size distributions to bimodal lognormal distributions for latitudinal bands spaced 15° apart. We compare their fitted size distributions for 30°-45°S, 45°-60°S and 60°-75°S to the annual zonal-mean size distributions for the DEFAULT_NoMSA case and each sensitivity case from the model. (There is no data available from Heintzenberg et al. (2000) for 75°S-90°S.) We note that changes in the aerosol size distributions between the measurement years and our simulated year (2014) are possible, even for these remote latitudes, and may result in apparent simulation errors and/or apparent model to measurement agreement biases.

The first and second Atmospheric Tomography Missions (ATom-1 and ATom-2) (https://espo.nasa.gov/missions/atom/content/ATom) took place from July 28 to August 22 of 2016 and January 26-February 22 of 2017, respectively. Carrying a comprehensive gas and particle chemistry payload, the NASA DC-8 aircraft systematically sampled the remote atmosphere, profiling continuously between 0.2 and 12 km. The data for both missions is publicly available (Wofsy et al, 2018). As a part of the instrumentation on board, a highly customized Aerodyne high-resolution time-of-flight aerosol mass spectrometer (AMS in the following; DeCarlo et al, 2006; Canagaratna et al, 2007) continuously measured the composition of submicron ($PM_1$), non-refractory aerosol at 1 Hz time resolution. The principle of operation and instrument/aircraft-operation specifics have been described in detail elsewhere (Dunlea et al., 2009; Kimmel et al., 2011; Schroder et al., 2018; Nault et al., 2018) and only the aspects specific to MSA quantification are discussed here.

The instrument flew in the same configuration for all four ATom missions, concluding in May 2018. Overall sensitivity (as determined daily from the ionization efficiency of nitrate, $IE_{NO3}$), relative ionization efficiencies and particle transmission (all determined periodically in the field) were stable over all four deployments. Particle phase MSA concentrations for all ATom flights are reported based on the intensity of the highly specific marker ion $CH_3SO_2^+$ (Phinney et al, 2006, Zorn et al, 2008). The quantification of MSA $PM_1$ concentrations from the signal intensity of the $CH_3SO_2^+$ fragment is described in detail in the SI, Sect. S5. Positive Matrix Factorization (Paatero 1994; Ulbrich et al., 2009) of the ATom-1 organic aerosol (OA) and sulfate data confirmed the specificity of the marker ion for MSA and the consistency of the field mass spectra with those acquired in the MSA calibrations. Importantly, it also confirmed that the AMS response to MSA is independent of the aerosol acidity, which varied significantly over the range of conditions found in ATom. Further details are provided in Sect S5.

For the data presented here, the AMS raw data was processed at 1 minute resolution. Under those conditions, the detection limit of MSA was in the range 1.5-3 ng sm$^{-3}$ (0.3-0.6 pptv), and will decrease with the square root of the number of averaged 1-minute data points. The uncertainty in the MSA quantification as detailed in the SI, Sect. S5, is comparable to that of sulfate, hence the overall uncertainty in the quantification is estimated to be +/-35% (2 standard deviations; Bahreini et al., 2009).





We compare our sensitivity simulations to the ATom data as follows: we subtract the DEFAULT_NoMSA sulfate mass (that accounts for sulfate and sulfuric acid from DMS/$SO_2$ oxidation but not MSA) for the months of August (ATom-1) and February (ATom-2) from the sulfate mass for the months of August and February for each sensitivity case that includes MSA for each grid box. The resultant differences in sulfate mass represents the model-predicted contributions of MSA to the

total sulfur budget for each case. We then compare the measured and predicted MSA mass by first averaging every ATom data point that falls within a given GC-TOMAS grid box. We then compare each averaged data point to that model grid box. The ATom data used in our analysis lies within 150-180° W (the Pacific ocean basin) and 10-40° W (the Atlantic ocean basin) and thus we use zonal averages of these longitude bands for both the ATom data and the GC-TOMAS output. We note that comparing monthly mean simulated values from 2014 to airborne measurements from a single point in time in 2016

and 2017 contributes to the apparent simulation errors.

To evaluate model performance, we calculate the log-mean bias (LMB), the slope of the log-log regression ($m$), and the coefficient of determination ($R^2$) between each cosampled GC-TOMAS grid box and averaged measurement point that falls within that GC-TOMAS grid box. The LMB is calculated through:

$$LMB = \frac{\sum_i^N (log_{10}(S_i) - log_{10}(O_i))}{N},$$  (3)

where $S_i$ and $O_i$ are the simulated and observed MSA masses, respectively, for each data point $i$, and $N$ is the number of data points. A LMB of 1 means that on average, the model overestimates the measurements by a factor of $10^1$ (10); a LMB of -1 means that on average, the model underestimates the measurements by a factor of $10^{-1}$ (0.1); a LMB of 0 indicates no bias between the model and measurements ($10^0 = 1.00$). LMB, $m$, and $R^2$ are summarized in Table 4. Since MSA is observed only in the particle-phase in the ATom measurements, we do not include the NoDMS_NoMSA (no DMS emissions in the

model) sensitivity case in our analysis of the ATom data. We present the aggregated results of the two campaigns, as well as results for each campaign and ocean basin. The ATom-1 mission provided more data points than the ATom-2 missions (1258 vs. 1000) and thus the aggregate results are slightly skewed towards the ATom-1 results.

**2.6 Study caveats**

This study is intended to examine the sensitivity of the aerosol size distribution and radiative impacts implied by the various sensitivity treatments of MSA (Table 2). However, our treatments of DMS and MSA still fall short of what is currently known about organic condensational behavior as well gas-phase and aqueous-phase oxidation pathways of DMS currently not included in GEOS-Chem. Assuming idealized semivolatile condensation with no re-evaporation due to conditional changes (e.g. change in temperatures, RH) may overestimate the amount of MSA able to condense on particles;

but it may also underestimate particle-phase MSA if conditions for condensation switch from unfavorable to favorable after MSA chemical production. Further, relying on E-AIM simulations to construct our volatility parameterization could have hidden biases due to an incomplete understanding of the system. Regarding chemistry uncertainties, the standard GEOS-Chem model does not include DMS oxidation through the OH or halogen addition pathways to dimethylsulfoxide (DMSO).



DMSO chemistry reduces the yield of sulfate formation from DMS/SO$_2$ oxidation (Breider et al., 2014) by increasing the yields of both gas-phase and aqueous phase MSA as well as aqueous-phase dimethyl sulfone (DMSO$_2$), another stable oxidation product (Hoffmann et al. 2016). To reduce the number of parameters for this study, we do not include the DMSO pathway. We acknowledge that neglecting this pathway will slightly bias our estimates of the contribution of sulfate and

MSA mass high and low, respectively. Further, aqueous-phase production of MSA would condense on CCN-sized particles, similar to aqueous phase sulfate (Sect 2.1), shifting the size distribution to larger sizes. Heterogeneous oxidation may limit the lifetime of MSA in the particle phase (Mungall et al., 2017; Kwong et al., 2018) although the reactive uptake coefficients from these studies are somewhat dissimilar, indicating a need for further study of the system. Regardless, neglecting heterogeneous chemistry could overestimate the estimate of the contribution of MSA to aerosol mass. Finally, if MSA does

participate in nucleation, it is unlikely that it will behave exactly like sulfuric acid, as it is treated here. All of the limitations described above are important and require further testing in detailed chemical models and chemical-transport models in order to determine their effects.

Another limitation of this study is our reliance upon the current ammonia inventory in GEOS-Chem as well as our cutoff value of 10 ppt of ammonia between the no ammonia and excess ammonia regimes (Sect. 2.2). Uncertainties in the

ammonia inventories over the oceans could change our results, as could a different cutoff value. As this study is focused on MSA sensitivities, we will leave sensitivities of MSA to ammonia for a future study. It is important to note that other bases such as amines could also have an important effect on MSA's effective volatility. However, the standard GEOS-Chem currently does not account for gas-phase bases beyond ammonia, and this sensitivity will also be left for a future study.

**3 Results and Discussion**

Figure 2 shows the global annual mean percent change (at 900 hPa and zonally) for submicron mass by adding MSA for the PARAM_NoNuc, ELVOC_NoNuc, SVOC_NoNuc, PARAM_Nuc, and ELVOC_Nuc simulations. Figure 3 shows the global annual mean percent change in N3 and N80 due to addition of MSA at 900 hPa and zonally for all model levels for each of these cases, and Fig. 4 shows the corresponding global annual AIE and DRE of MSA. Figure 5 shows the

global annual mean percent contribution from DMS/SO$_2$ oxidation (at 900 hPa and zonally) alone (not including MSA) to submicron mass, N3, N80, AID, and DRE. Figure 6 and Table S3 summarises the results of Figs. 2, 3, 4, and 5. All of the numerical statistics presented in Sects. 3.1-3.4 are for the annual mean, either globally or between 30°-90°S. Each case with MSA is analyzed for the change relative to DEFAULT_NoMSA to determine the impact that MSA has on the size distribution and resulting radiative effects (positive values indicate that the inclusion of MSA increases a given metric). For

reference, Figure S7 provides the absolute number concentration for N3 and N80 at 900 hPa and zonally for all model levels for the DEFAULT_NoMSA simulation. We will refer back to these figures in the following sections.

**3.1 Volatility-dependent impact of MSA if MSA does not participate in nucleation**





The top rows of Figs. 2 and 3 show the global annual mean percent change at 900 hPa and zonally from adding MSA using the volatility parameterization without nucleation (PARAM_NoNuc - DEFAULT_NoMSA) for submicron aerosol mass (Fig. 2) and N3 and N80 (Fig. 3). By adding MSA with these assumptions, we predict at 900 hPa an increase in submicron mass of 0.7% globally and 1.3% between 30°S-90°S; a decrease in N3 of -3.9% globally and -8.5% between

30°S-90°S; and an increase in N80 of 0.8% globally and 1.7% between 30°S-90°S (Fig. 6 and Table S3). These MSA impacts are limited by ammonia availability. Figures S1 and S2 show that many oceanic regions are predicted to have annual and seasonal ammonia mixing ratios of less than 10 ppt. Below 10 pptv of ammonia, MSA condensation as SVOC-like or VOC-like (no condensation) (Fig. 1a) and MSA condensation will only be SVOC-like if the RH > 90%; under these conditions for the majority of the year, MSA will be a VOC-like species over Antarctica (low RH conditions) and often an

SVOC-like species over the southern-ocean boundary layer (high RH conditions). Only in the Southern Hemisphere (SH) winter months does ammonia exceed 10 ppt over appreciable regions in the southern oceans (Fig. S2); during this time, MSA condensation is ELVOC-like due to cold temperatures (Fig. 1b). As shown in D'Andrea et al. (2013), ideal-SVOC material largely condenses primarily to accumulation-mode particles, which in turn suppresses N3 through increased coagulation and reduced nucleation and has little impact on N80. In the midlatitudes, the annual and seasonal ammonia

concentrations often exceed 10 ppt, and thus MSA condensation will be either ELVOC-like under low-temperature and/or high-RH conditions or SVOC-like under high-temperature and/or low-RH conditions. D'Andrea et al. (2013) showed that adding ELVOC material can increase N80 by increasing growth of ultrafine particles but also can suppress N3 through the same coagulation/nucleation feedbacks. This combination of ammonia-rich and ammonia-poor regions lead to MSA giving an overall weak increase in N80 with a large suppression of N3 in some regions. We note that these results are somewhat

sensitive to the simulated ammonia concentrations and may be sensitivity to the ammonia cutoff of 10 ppt in the MSA-volatility parameterization. As there are already uncertainties in many other dimensions, we do not attempt to quantify the sensitivity of MSA towards ammonia in this work.

The idealized volatility cases, ELVOC_NoNuc (Figs. 2 and 3, second row) and SVOC_NoNuc (Figs. 2 and 3, third row) help to highlight and further explain MSA's volatility-dependent contribution towards growth. In both of these cases,

100% of the formed MSA goes to the particle phase, unlike with the MSA volatility parameterization, where MSA may not condense in the absence of base at lower RHs. Hence, the global annual MSA mass is nearly double in these cases compared to when using the parameterization (Table 2; Fig. 2). The addition of MSA in ELVOC_NoNuc allows for an increase in condensable material that condenses to the Fuchs-corrected surface area through ELVOC-like condensation, which increases the growth rate of all particle sizes. Conversely, MSA in SVOC_NoNuc allows for an increase of SVOC-like material that

will condense preferentially to larger particles through SVOC-like condensation (but still irreversible condensation). In both the ELVOC_NoNuc and SVOC_NoNuc cases, N3 concentrations are reduced due to increased coagulational losses and decreased nucleation rates because of the added MSA mass (D'Andrea et al., 2013). When MSA condensation is treated as ELVOC-like, the smaller particles grow more quickly into the larger sizes, so N80 increases by 9.1% globally and by 22.2% between 30°S-90°S at 900 hPa (Fig. 6 and Table S3). When MSA condensation is instead treated as SVOC-like, the largest



particles uptake MSA preferentially to smaller particles, and the N80 are not greatly impacted by the addition of MSA. (The slight boost in N80 for SVOC_NoNuc in the tropical upper troposphere (UT) is due to the very low accumulation-mode concentration in this region: the SVOC material condenses to ultrafine particles in this region.)

The changes in DRE and AIE resulting from the addition of MSA for these three no-MSA-nucleation cases (Fig. 4, top three rows) depend roughly on the changes in N80 (the activation diameter for determining CDNC will depend on local particle hygroscopicity and concentrations). The DRE generally scales linearly with aerosol mass (Fig. 2, top three rows). As MSA is assumed to have the same properties as sulfate, which is is assumed to be purely scattering, any increases in MSA mass results in a negative radiative effect. However, the DRE also depends on aerosol size; the scattering efficiency peaks between ~300-900 nm, depending upon the aerosol composition and shape (Seinfeld and Pandis, 2016, their Fig. 15.8). The

change in DRE when MSA is included using the volatility parameterization (PARAM_NoNuc) is less negative than that of ELVOC_NoNuc and SVOC_NoNuc at -15 mW m$^{-2}$ globally (-26 mW m$^{-2}$ between 30°S-90°S), because the parameterization yielded a smaller mass increase than the ideal volatility simulations. ELVOC_NoNuc and SVOC_NoNuc have almost identical changes in submicron aerosol mass (Fig. 6; Table S3) but the DRE is -25 mW m$^{-2}$ globally (-44 mW m$^{-2}$ between 30°S-90°S) for SVOC_NoNuc and -0.02 W m$^{-2}$ globally (-34 mW m$^{-2}$ between 30°S-90°S) for ELVOC_NoNuc

(Fig. 6; Table S3). MSA will preferentially condense to larger aerosol when its condensation is SVOC-like, and so even though ELVOC_NoNuc shows a larger increase in N80, SVOC_NoNuc increases the fraction of particulate mass in the peak scattering efficiency regime.

       The AIE instead scales the with aerosol number concentration of particles large enough to act as CCN: PARAM_NoNuc's AIE (-8.6 mW m$^{-2}$ globally, -17 mW m$^{-2}$ between 30°S-90°S) reflects the small increase in N80 (0.8%

globally and 1.7% between 30°S-90°S at 900 hPa) (Fig. 6; Table S3). The larger increase in N80 for ELVOC_NoNuc results in the larger cooling tendency in the AIE, at -0.075 W m$^{-2}$ globally (-150 mW m$^{-2}$ between 30°S-90°S), and the slight decrease in N80 for SVOC_NoNuc results in the slight warming tendency in AIE at 7.5 mW m$^{-2}$ globally (11 mW m$^{-2}$ between 30°S-90°S) (Fig. 6; Table S3).

       These annual results show in Fig. 6 and Table S3 that if MSA does not take part in nucleation, the submicron

aerosol mass will increase, causing a cooling tendency in the DRE, and N3 will decrease regardless of the volatility assumed. However, the changes in N80 are sensitive to the volatility assumption and will only increase if MSA condensation is ELVOC-like at least over some spatial and temporal scales, thereby causing a further cooling tendency in the AIE.

**3.2 Volatility-dependent impact of MSA if MSA does participate in nucleation**

To test the potential influence on aerosol size distributions if MSA contributes to nucleation, we model allow MSA to participate in binary and ternary nucleation with the same efficacy as sulfuric acid. This provides an upper bound in the potential contribution of MSA towards nucleation (at least for the nucleation schemes tested here). Figures 2, 3, and 4 (fourth rows) show the global annual mean percent changes between DEFAULT_NoMSA and PARAM_Nuc. MSA will have the same effective volatility as discussed for PARAM_NoNuc (Sect. 3.1) but will now participate in nucleation under ELVOC-





like regimes. For PARAM_Nuc, we can clearly see that the when the ammonia concentrations reach above 10 ppt in the SH winter months over the Southern Ocean (Fig. S4), MSA acts as an ELVOC-like species and contributes strongly to nucleation in these sulfuric-acid poor regions. The addition of MSA in ELVOC_Nuc has the largest impact on N3, N80, and the AIE of any of our cases with an increase in N3 of 153.4% globally (397.7% between 30°S-90°S), an increase in N80 of

23.8% globally (56.3% between 30°S-90°S), and a decrease for the AIE of -0.18 W m$^{-2}$ globally (-0.39 W m$^{-2}$ between 30°S-90°S). MSA in PARAM_Nuc also has a large increase in N3 (112.5% globally and 309.9% between 30°-90° Sat 900 hPa) but only increase N80 by 2.1% globally (4.4% between 30°-90° S), again indicating that MSA often undergoes SVOC-like or ELVOC-like condensation volatility parameterization.

The increase in N80 from MSA in PARAM_Nuc is about double that of the increase from MSA in

PARAM_NoNuc, and the change in AIE is similarly slightly double for PARAM_Nuc. The global annual changes in submicron mass and the DRE is quite similar between the two PARAM cases. However, N80 increases more over the northern hemisphere (NH) high latitude ocean regions for PARAM_Nuc than for PARAM_NoNuc, and as a result, the northern oceans experience a stronger regional negative AIE when MSA is allowed to participate in nucleation. As noted in Sect. 3.1, there are uncertainties from the ammonia concentrations and cutoff point of 10 ppt for PARAM_Nuc, but we will

not attempt to quantify them here.

These results indicate that if MSA does participate in nucleation, the largest climate-relevant change is anticipated to be an increased cooling tendency for the AIE as compared to if MSA does not participate in nucleation. The change in DRE will be similar though, as MSA mass is not predicted to significantly change between non-nucleating and nucleating cases. This study provides an upper bound on the contribution of MSA to nucleation: if MSA is less efficient at nucleating

than sulfuric acid, it is present in relatively sulfuric-acid poor regions and would still be able to increase N3 concentrations (although possible by less than predicted here). Microphysical feedbacks (increased condensation and coagulation sinks from increased N80) will then limit the effect that small changes in N3 can have on N80 and radiative effects.

### 3.3 Comparison of MSA impacts to the contribution from SO$_2$ formed in DMS oxidation

By removing DMS from the simulation entirely (NoDMS_NoMSA case; Figs. 5 and 6 and Table S3), we determine the baseline contribution of the simulated sulfuric acid and sulfate from DMS/SO$_2$ oxidation to the aerosol size distribution in GEOS-Chem-TOMAS at 900 hPa. The sulfate and sulfuric acid from DMS/SO$_2$ oxidation provides larger changes in submicron mass and N80 than MSA does in any of our sensitivity cases. The contribution of SO$_2$ from DMS to submicron mass is 4-6 times that of the MSA contribution. However, about ⅔ of this mass increase from DMS/SO$_2$ comes through

aqueous oxidation of SO$_2$ to sulfate, which adds mass (but not number) to already-CCN-sized particles (Pierce et al., 2013) suppressing nucleation and growth. The remaining ~⅓ of the mass comes from gas-phase formation of sulfuric acid, which nucleates particles and condenses irreversibly to the Fuchs-corrected surface area, potentially increasing the number of CCN-sized particles. Overall, N3 and N80 increase due to the inclusion of the DMS/SO$_2$ pathway (N3 by 7.3% and N80 by 12.2% globally and N3 by 19.5% and N80 by 24.3% between 30°S-90°S at 900 hPa). The increases in both N3 and N80 are




strongly damped by the formation of aqueous sulfate. The changes in N3 at 900 hPa indicate the relative importance of the sulfuric acid produced by DMS/SO₂ oxidation for nucleation compared to other sources of sulfuric acid. N3 generally increases in remote regions where sulfuric acid from DMS/SO₂ oxidation would be the main source of sulfuric acid. There are also regions of decrease in N3 in remote regions: the condensation and coagulation sinks increase from aqueous sulfate

formation, and in some regions this competition effectively scavenges N3 faster than $H_2SO_{4\,DMS}$ forms new particles. Because of the large increase in submicron mass from the sulfuric acid and sulfate from DMS/SO₂ oxidation, the DRE from DMS/SO2 is -120 mW m⁻² globally (-173 mW m⁻² between 30°S-90°S), about 5 times larger than MSA for any of our assumptions. On the other hand, the AIE cooling tendency of -46 mW m⁻² globally and -38 mW m⁻² between 30°S-90°S, was within the range of AIEs from MSA that we predicted, which is due to the N80 damping of DMS/SO₂ due to aqueous sulfate

formation. Thus, overall we predict the DRE from MSA to be at least 5 times weaker than from DMS/SO₂, but the AIE may be of similar magnitude depending on the properties of MSA.

**3.4 Analysis of model-measurement comparisons**

Figure 7 shows the comparison between the annual zonal-mean particle number size distributions compiled in

Heintzenberg et al. (2000; hereon referred to as Heintzenberg) and the GC-TOMAS simulated annual-mean particle number size distributions within the boundary layer for the latitude bands of 30°S-45°S, 45°S-60°S, and 60°S-75°S (no data was provided in Heintzenberg between 75°S-90°S). We focus this comparison to the southern oceans region as this region has the strongest influence from DMS and its oxidation products. It is also less likely to be influenced by changing anthropogenic emissions that may have occurred between the time of the measurements compiled in Heintzenberg (between ~1970-1999)

and 2014 (the year of the model run) than higher latitudes (e.g. Pierce and Adams, et al., 2009a; Gordon et al., 2017). We see that all model simulations underpredict both the Aitken and accumulation modes of Heintzenberg, but that the simulations that allow MSA to participate in nucleation (ELVOC_Nuc and PARAM_Nuc) give the best model-to-measurement agreements for the Aitken mode for each latitude band, with ELVOC_Nuc performing the best across the model cases. Further, ELVOC_Nuc shows the highest number of particles in the accumulation mode, particularly between 60°-75° S.

These results point to the necessity of another source of ultrafine particles over the southern oceans than is being currently accounted for in the model. These particles may be produced locally from ultrafine sea spray (Pierce and Adams, 2006), local nucleation (not necessarily through MSA), or entrainment of ultrafine particles from the free troposphere (Clarke et al., 2002).

For the ATom mission, Figure 8 provides 1:1 plots for each sensitivity case's predicted MSA mass versus the

observed MSA mass from the aggregated ATom campaigns. Each subplot provides the LMB, $m$, and $R^2$ statistics for the given sensitivity case. LMB, $m$, and $R^2$ statistics are also provided for each campaign and ocean basin in Figs. S7-S10; Figures S11-S14 show the zonally averaged simulated MSA concentrations for each basin and campaign with the corresponding particle-phase MSA measurements overlaid. Figure 8 indicates that for the aggregated campaigns, the model cases in which MSA always condenses to the particle phase (the SVOC_NoNuc, ELVOC_NoNuc, and ELVOC_Nuc cases)





overpredict MSA mass, with positive LMBs between 0.27 and 0.3 (overpredictions of a factor of 1.9-2). The PARAM_NoNuc and PARAM_Nuc cases do not allow MSA to condense to the particle phase under low-base/high-temperature/low-RH conditions (Fig.1). As a result, the PARAM cases instead slightly underpredict MSA mass, with LMBs of -0.1 and -0.08 (underpredictions by a factor of 0.79 and 0.83). Overall, when the parameterization is not used, too much

MSA mass is allowed to condense relative to the observations. Given the large improvement in LMB through the use of the parameterization (with roughly similar $R^2$ and $m$ values), we feel that these results support the use of the volatility parameterization of MSA.

The $R^2$ values are quite low across cases, with the cases with the parameterization giving the highest $R^2$ values, at 0.09. The $m$ values are similarly low, with the SVOC_NoNuc, ELVOC_NoNuc, and ELVOC_Nuc cases giving the highest

$m$ values, at 0.33-0.34. However, we are comparing monthly grid-box mean model predictions to individually grid-box averaged measurements taken during a different year than the simulation year. Further, using monthly mean model predictions on the y-axis (Fig. 8) decreases variability, which reduces the slope. These considerations contribute to lower values of $R^2$ and $m$.

The Heintzenberg and ATom model-measurement comparisons disagree on which MSA assumptions lead to the

best performance in GC-TOMAS. However, the Heintzenberg analysis considers number size distribution whereas the ATom analysis considers total particle-phase MSA mass. The model-measurement improvement for the Heintzenberg study is most strongly seen within the Aitken mode (the smallest reported particle sizes). Aitken-mode-sized particles contribute little to total mass compared to larger particles. Further, it is not possible to determine from this study whether the source of ultrafine particles that could explain the size of the Aitken modes in Heintzenberg comes from MSA another primary or

secondary source. On the other hand, the ATom comparison suggests that using the MSA volatility parameterization helps predict the MSA mass concentrations more accurately.

**4 Conclusions**

We used the GEOS-Chem chemical transport model coupled to the TOMAS aerosol microphysics module to test

the sensitivity of the aerosol size distribution and resulting changes in the direct and indirect effects to the condensational and nucleating behavior of methanesulfonic acid (MSA), an oxidation product of dimethylsulfide (DMS). GEOS-Chem-TOMAS (GC-TOMAS) normally simulates sulfuric acid and sulfate from DMS/SO$_2$ oxidation but does not include MSA within the size-resolved portion of the model; we used this setup as our default model case (DEFAULT_NoMSA). We considered both the global annual mean size distributions and the annual mean in the southern oceans regions (30S°-90°S) at

900 hPa for each sensitivity case compared to DEFAULT_NoMSA. We further evaluated the model output against two different measurement sets: zonal-mean number size distributions compiled from ship-based measurements taken in the southern oceans and particle-phase MSA mass concentrations obtained from aircraft data over the Atlantic and Pacific ocean basins for the months of August and February.





As the effective volatility of MSA is uncertain, we used the Extended Aerosol Inorganics Model (E-AIM) to build a parameterization for GC-TOMAS of MSA's potential volatility as a function of temperature, relative humidity, and available gas-phase base. For simplicity, we only allowed MSA to condense as ideally nonvolatile or semivolatile, or to be volatile and not condense at all under the parameterization. If MSA was ideally nonvolatile, it contributed to the size distribution through

condensation proportional to the Fuchs-corrected aerosol surface area distribution (effectively nonvolatile or ELVOC-like condensation). If MSA was instead ideally semivolatile, it contributed to the size distribution through condensation proportional to the aerosol mass distribution (quasi-equilibrium or SVOC-like condensation). Regardless of the volatility treatment, condensed MSA was not allowed to evaporate back to the gas-phase, as gas-phase MSA was not explicitly tracked in the model. Along with the parameterization, we tested limiting volatility cases, allowing MSA to only be ELVOC-like or

SVOC-like. We also performed separate simulations in which MSA could participate in nucleation, using both the MSA volatility parameterization and the ELVOC-like and SVOC-like MSA assumptions. (MSA participated in nucleation only when it was under ELVOC-like conditions in the parameterization; it always participated in nucleation in the ELVOC simulation). When using the volatility parameterization, including MSA in the model changed the global annual averages of submicron aerosol mass by 1.2%, N3 by -3.9% (non-nucleating) or 112.5% (nucleating), N80 by 0.8% (non-nucleating) or

2.1% (nucleating), the aerosol indirect effect by -8.6 mW m$^{-2}$ (non-nucleating) or -26 mW m$^{-2}$ (nucleating), and the direct radiative effect by -15 mW m$^{-2}$ (non-nucleating) or -14 mW m$^{-2}$ (nucleating). Across all simulations, including MSA in the model changed the global annual averages of submicron aerosol mass by 0.7% to 1.2%, N3 by -3.9% to 153.4%, N80 by -0.2% to 23.8%, the aerosol indirect effect by -0.18 W m$^{-2}$ to 0.0075 W m-2, and the direct radiative effect by -25 mW m$^{-2}$ to -13 mW m$^{-2}$, depending on the assumed volatility and nucleating ability of MSA.

The contribution from the sulfuric acid and sulfate from DMS/SO$_2$ oxidation to the submicron aerosol mass is 4-6 times that of the contribution from DMS/MSA, leading to a global cooling from the DRE 5-10 times that of MSA, at -120 mW m$^{-2}$. However, because much of the aerosol mass from DMS/SO2 is added through aqueous sulfate formation, which suppresses nucleation and growth, the changes in N3, N80, and the AIE from DMS/SO$_2$ oxidation products are smaller and on the order of changes in these metrics from including MSA in the model.

The model-measurement annual zonal number size distribution comparisons to the ship-based measurements compiled in Heintzenberg et al. (2000) of the southern-ocean region (Fig. 11) show an underprediction of the Aitken mode across cases, with the best agreement in the Aitken mode coming from the cases that allow MSA to act as a nucleating nonvolatile compound (ELVOC_Nuc and PARAM_Nuc). These results indicate the necessity of another source of ultrafine particles over the southern oceans that is currently not being accounted for in the model. However, it is not possible to

conclude based on this study where the source of extra ultrafines is coming from. More studies over the oceans detailing the chemical compositions of the smallest particle sizes are needed in order to help determine the origins of nucleating material in these remote regions.

The model-measurement comparisons of total particle-phase MSA mass from the aircraft data taken during the ATom-1 and ATom-2 campaigns compared to the predicted mean MSA mass indicate that PARAM_Nuc and



PARAM_NoNuc cases perform the best, and that the cases in which MSA is always allowed to condense to the particle phase overpredict MSA mass. As the Heintzenberg and the ATom model-measurement comparisons are based on dissimilar metrics (number size distribution versus particle-phase MSA mass) over dissimilar spatial extents (surface-based ground and ship measurements versus aircraft measurements continuously profiling between 0.2 and ~13 km), we cannot definitively

state that any one sensitivity case appears to best-fit both the Heintzenberg and ATom measurements. Along with these model-measurement comparisons, we provided a detailed description of the calibration for detecting MSA applied to the Aerodyne high-resolution time-of-flight aerosol mass spectrometer (AMS) present during the ATom campaigns in the supplement as a reference for the AMS community.

As there are uncertainties in both MSA's behavior (nucleation and condensational) and the DMS emissions

inventory, further modelling and measurement studies on both fronts are needed to better constrain MSA's current and future impact upon the global aerosol size distribution and radiative effect. Under the simulation tested in this work, MSA tends to have small ($< -0.1$ W m$^{-2}$) global annual radiative effects (DRE and AIE); in general, the forcings are predicted to be cooling effects. The contributions to the size distribution and radiative effects increase in magnitude in the southern oceans, where MSA concentrations are highest and more-pristine conditions exist. Although small, the radiative effects from MSA and the

associated size distribution dependencies should be well-characterized to more-fully understand the role of changing DMS emissions in a changing climate. This study provides a first look at some of these potential dependencies and indicates possible directions for future modelling and measurement studies.

**Data availability**

Data for the ATom campaigns is / will be posted publicly at https://doi.org/10.3334/ORNLDAAC/1581.

**Author contributions**

ALH, JRP, and BC defined the scientific questions and scope of this work. ALH and BC performed all GEOS-Chem model simulations and off-line calculations with help from JKJ, BC, and JRP. JRP performed the E-AIM calculations. PCJ, BAN,

JCS, and JLJ carried out the primary measurements and data processing for the ATom field campaign, as well as campaign supervision and design. ALH prepared the primary text with substantial contributions from JRP, JKJ, BC, PCJ, and JLJ. PCJ provided the detailed description provided in the supplement of the calibration method used for detecting MSA during the ATom field campaign, with additional contributions from JLJ.

**Competing interests**

The authors declare that they have no competing interests.

**Acknowledgements**



This research was supported by the US Department of Energy's Atmospheric System Research, an Office of Science, Office of Biological and Environmental Research program, under Grant No. DE-SC0011780 and by the U.S National Oceanic and Atmospheric Administration, an Office of Science, Office of Atmospheric Chemistry, Carbon Cycle, and Climate Program, under the cooperative agreement awards #NA17OAR430001. B.C. was supported under the Climate Change and

Atmospheric Research programme at 1164 NSERC, as part of the NETCARE project. The CU-Boulder group was supported by NASA NNX15AH33A and NNX15AJ23G.

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



**Tables**

**Table 1.** Fit coefficients for the MSA volatility parameterization equation.

| Variable | Value |
|----------|-------|
| $a$ | $2.52 \times 10^2$ |
| $b$ | $6.19 \times 10^{-1}$ |
| $c$ | $3.49 \times 10^{-2}$ |
| $d$ | $5.6 \times 10^{-4}$ |
| $e$ | $3.32 \times 10^{-6}$ |



**Table 2.** Description of simulations.

| Simulation | Description |
| --- | --- |
| DEFAULT_NoMSA | Default model simulation: MSA does not contribute to the particle size distribution in GEOS-Chem-TOMAS (GC-TOMAS). The default GC-TOMAS v10.01 DMS emissions are used, and $SO_2$, sulfate, and sulfuric acid from DMS does influence the particle size distribution. |
| PARAM_NoNuc (NoNuc = does not nucleate particles) | Parameterization for MSA from E-AIM simulations: volatility is based on $NH_3$, T and RH. MSA can act as non-volatile and non-nucleating, semivolatile, or volatile (no condensation). |
| ELVOC_NoNuc | MSA is assumed to be non-volatile and condenses proportionally to the surface area distribution. |
| SVOC_NoNuc | MSA is assumed to be semivolatile and condenses proportional to the mass distribution. |
| ELVOC_Nuc | Like ELVOC_NoNuc, but MSA acts like sulfuric acid in nucleation. |
| NoDMS_NoMSA | All DMS emissions are turned off in the model; all other parameters are the same as the DEFAULT_NoMSA case. |
| DEFAULT_NoMSA_Lana | Default case using the Lana et al. (2010) DMS emissions inventory. |
| DEFAULT_NoMSA_2xDMS | Default case with global DMS emissions increased by a factor of two. |
| PARAM_NoNuc_Lana | Use the settings of PARAM_NoNuc with the Lana et al. (2010) DMS emissions inventory. |
| PARAM_NoNuc_2xDMS | Increase DMS emissions by a factor of two, using the settings of PARAM_NoNuc |



**Figures**

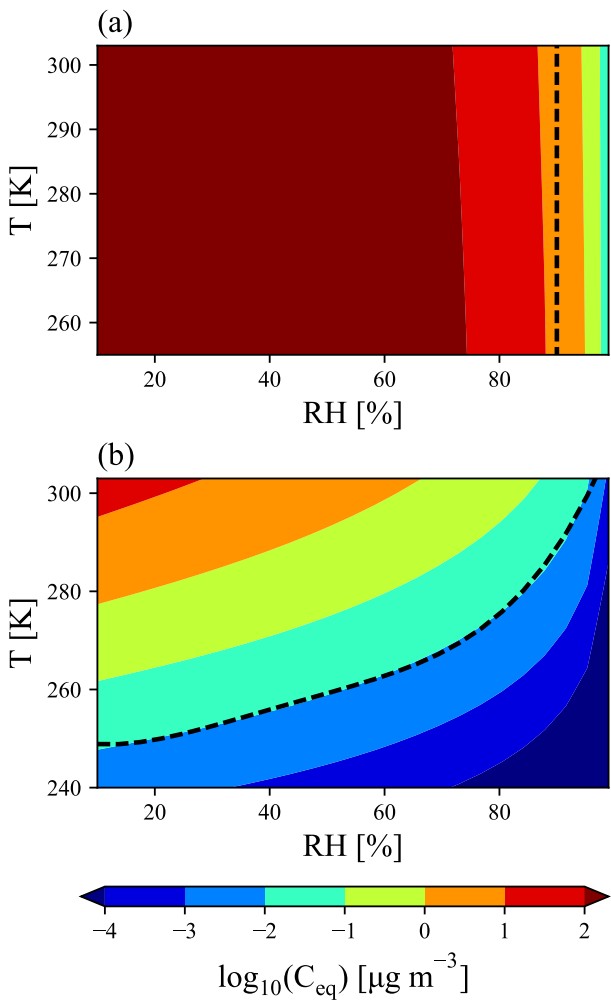

**Figure 1.** E-AIM prediction of MSA equilibrium vapor pressure above the particle mixture ($C_{eq}$) under conditions with (a)
no free ammonia and (b) high free ammonia (3 times as many moles of ammonia as MSA). (a) The dashed line at 90% RH
indicates the cut-off for representing MSA as a VOC-like (left of the line) or an SVOC-like (right of the line) species. (b)
The dashed line is described by Eq. 1 in the text. Above the dashed line, MSA is treated as an SVOC-like species; below the
dashed line, MSA is treated as an ELVOC-like species.



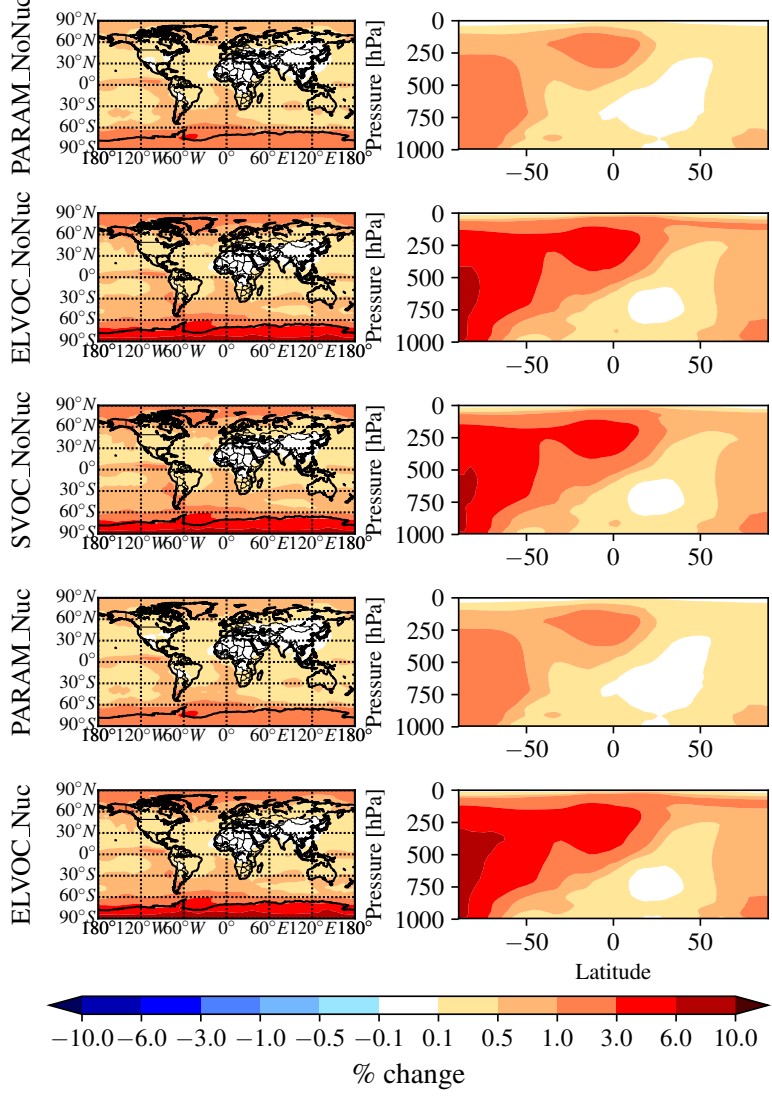

**Figure 2.** Global annual mean percent change in submicron aerosol mass due to the addition of MSA at 900 hPa (first column) and global zonal annual mean percent change (second column) between DEFAULT_NoMSA and PARAM_NoNuc (first row), ELVOC_NoNuc (second row), SVOC_Nuc (third row), PARAM_Nuc (fourth row), and ELVOC_Nuc (fifth row) (warm colors indicate an increase in submicron mass as compared to DEFAULT_NoMSA).





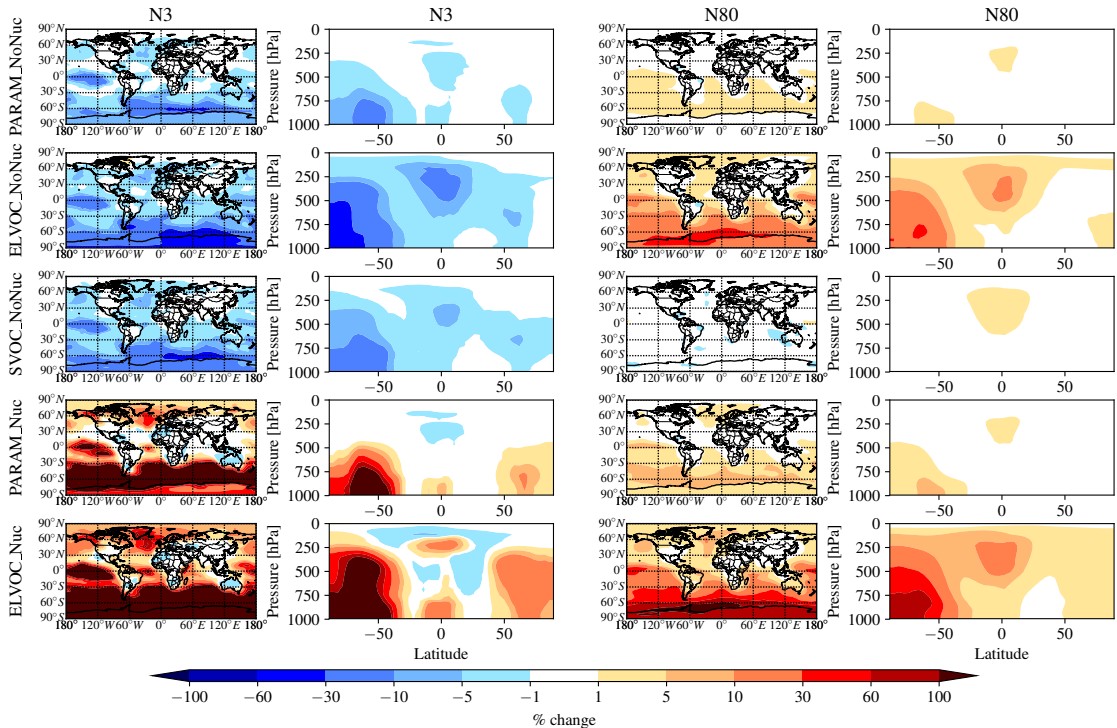

5  **Figure 3.** Global annual mean percent change in N3 and N80 at 900 hPa (first and third columns) and global zonal annual mean percent change (second and fourth columns) between DEFAULT_NoMSA and PARAM_NoNuc (first row), ELVOC_NoNuc (second row), SVOC_Nuc (third row), PARAM_Nuc (fourth row), and ELVOC_Nuc (fifth row) (warm colors indicate an increase in N3/N80 as compared to DEFAULT_NoMSA). First and second column: N3 (the number concentration of particles with diameters larger than 3 nm). Third and fourth column: N80.



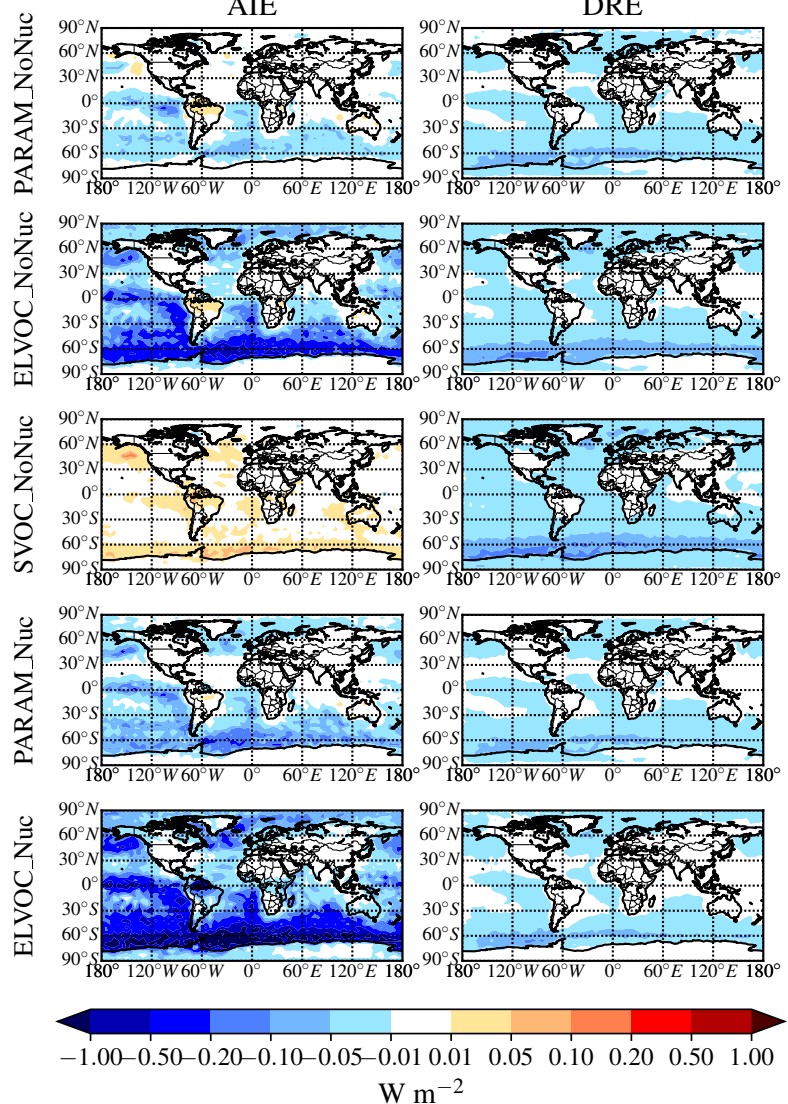

**Figure 4.** Global annual mean change in W m$^{-2}$ for the aerosol indirect effect (AIE; first column) and the direct radiative effect (DRE; second column) between DEFAULT_NoMSA and PARAM_NoNuc (first row), ELVOC_NoNuc (second row), SVOC_Nuc (third row), PARAM_Nuc (fourth row), and ELVOC_Nuc (fifth row) (warm colors indicate an increase in the AIE/DRE as compared to DEFAULT_NoMSA).





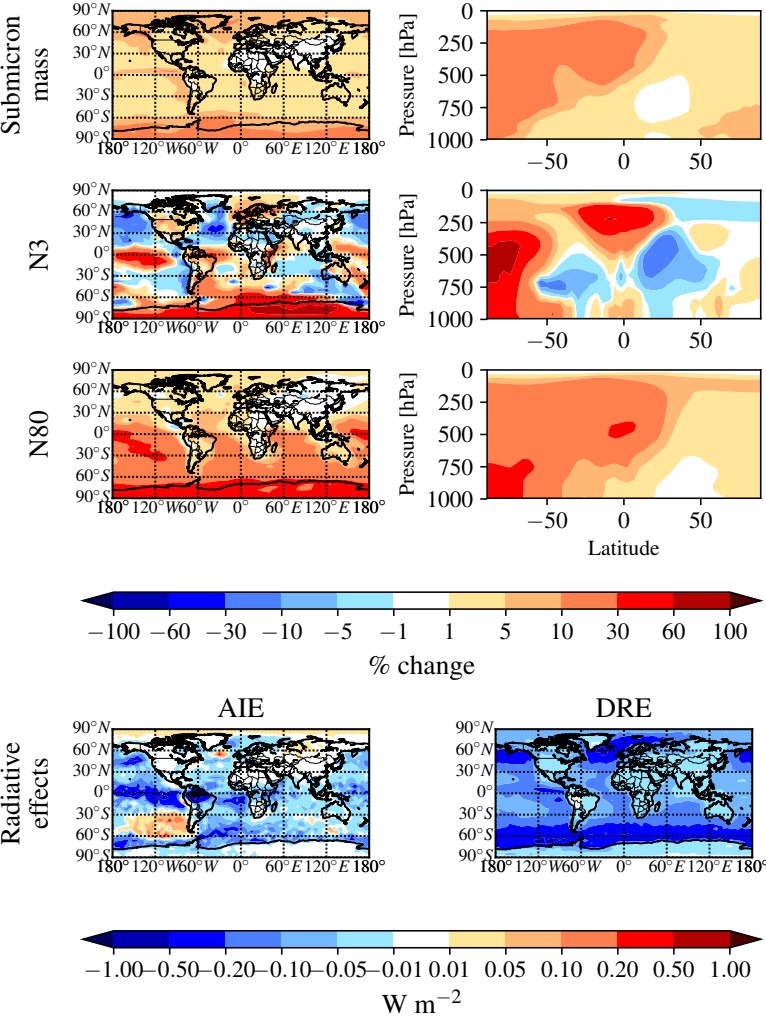

**Figure 5.** Global annual mean changes between the NoDMS_NoMSA and DEFAULT_NoMSA simulation. First row: percent change in submicron aerosol mass at 900 hPa (left) and zonally (right). Second row: percent change in N3 at 900 hPa (left) and zonally (right). Third row: percent change in N80 at 900 hPa (left) and zonally (right). Fourth row: change in W m$^{-2}$ in the radiative effects. This figures gives the contribution from sulfate and sulfuric acid produced from DMS/SO$_2$ oxidation to the aerosol mass, number, and radiative effects. Warm colors indicate that sulfate and sulfuric acid produced from DMS/SO$_2$ oxidation increase the metric.





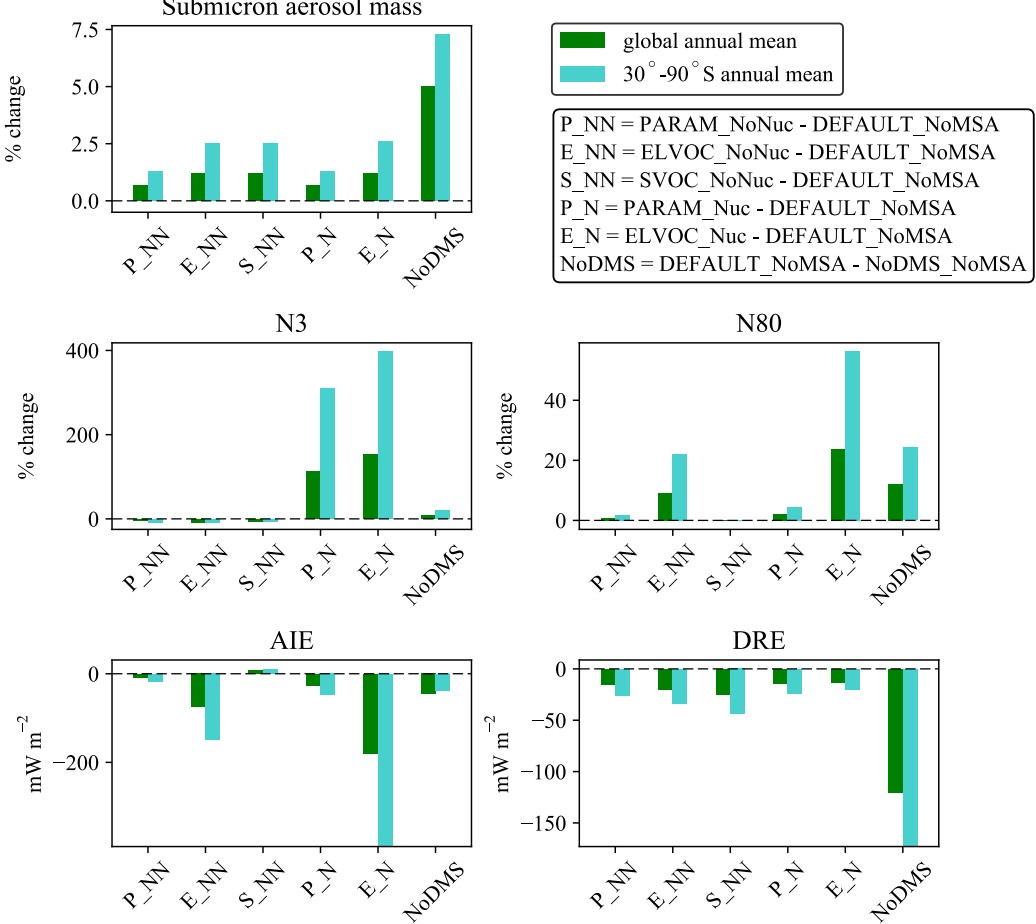

**Figure 6.** Annual mean changes due to MSA at 900 hPa for each MSA simulation relative to the DEFAULT_NoMSA simulation for submicron aerosol mass, N3, N80, all expressed as percent changes, and radiative forcing changes in AIE and DRE, both expressed as changes in W m$^{-2}$. Positive values for any metric for PARAM_NoNuc (P_NN), ELVOC_NoNuc (E_NN), SVOC_NoNuc (S_NN), PARAM_Nuc (P_N), and ELVOC_Nuc (E_N) all indicate that the addition of MSA increases that metric relative to the DEFAULT_NoMSA simulation. The DEFAULT_NoMSA-NoDMS_NoMSA (NoDMS) columns shows the contribution of the sulfate and sulfuric acid from DMS/SO$_2$ oxidation present in the DEFAULT_NoMSA simulation; positive values of a metric indicate that the sulfate and sulfuric acid increases that metric compared to a simulation with no DMS emissions. Numerical values for each bar are provided in Table S3.



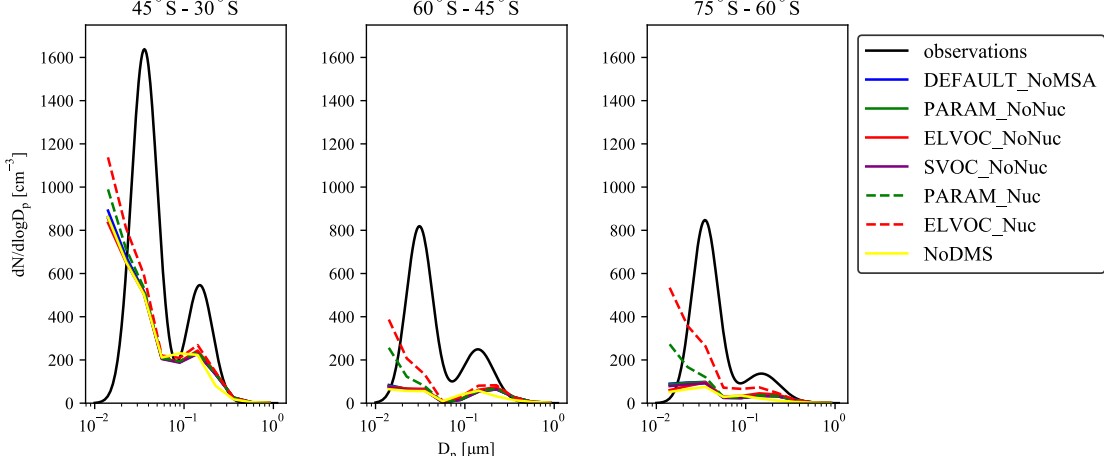

**Figure 7.** Comparison of simulated annual mean particle number size distributions to the annual zonal particle number size distributions compiled in Heintzenberg et al. (2000) (black lines) for the southern oceans. No data was available in Heintzenberg et al. (2000) for 75-90° S. We match the grid boxes sampled in their study to the GEOS-Chem-TOMAS grid boxes; due to sparseness of data, we do not attempt to discuss seasonal variabilities in this comparison.

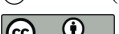



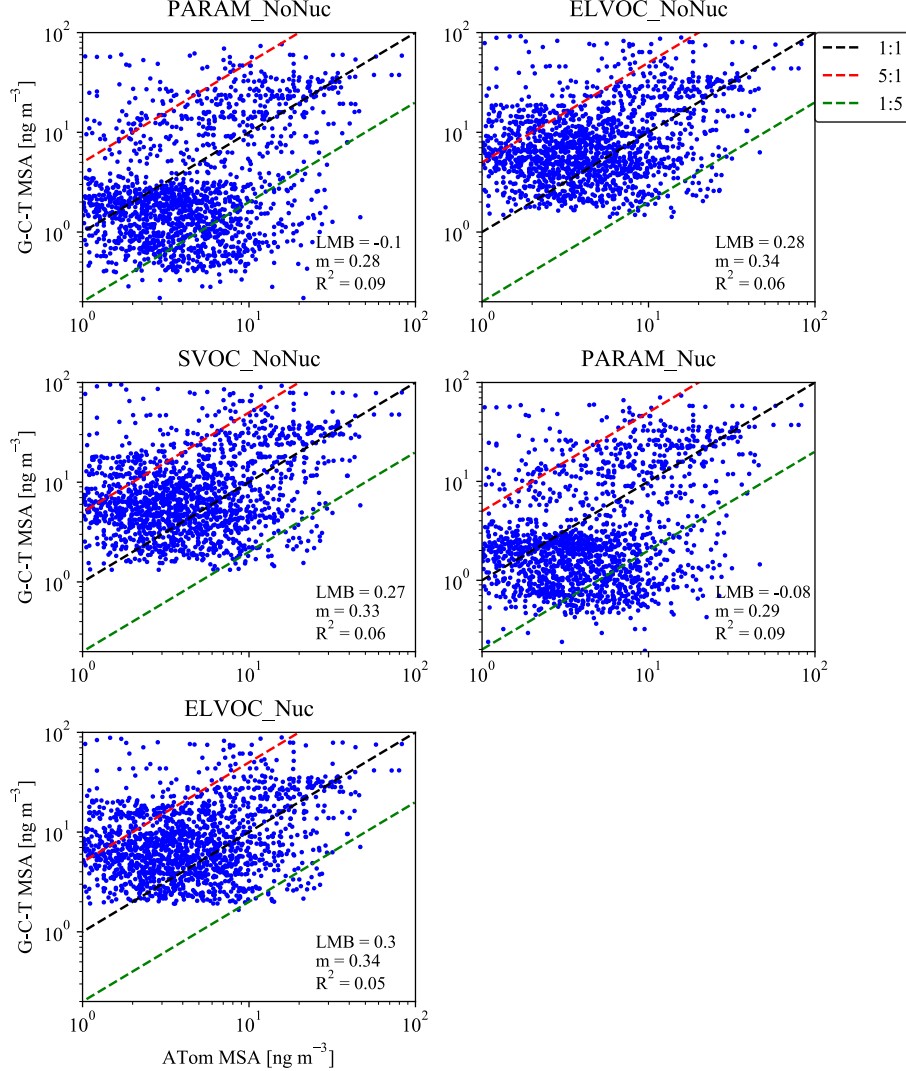

**Figure 8.** 1:1 (black dashed line) plots for the simulated mean MSA mass for the months of August/February and measured MSA mass during the ATom-1/Atom-2 campaigns (July 28-August 22 2016 / January 26-February 22 2017). Each subpanel gives the calculated log-mean bias (LMB), slope (*m*), and coefficient of determination ($R^2$) between the ATom data and the sensitivity simulation. The red and green dashed lines indicate 5:1 and 1:5 lines. Simulated MSA mass is calculated by subtracting the total sulfate mass for the base case from each sensitivity case.