# Peer review of "The potential role of methanesulfonic acid (MSA) in aerosol formation and growth and the associated radiative forcings"

_Atmospheric Chemistry and Physics, 2018_

## Referee Comment (RC1) · Anonymous Referee #2 · 10 Jan 2019

The manuscript by Hodshire et al uses a global aersol model to quantify the role of MSA in the aerosol-climate system. The role of MSA has not been quantified previously, with the majority of current aerosol and climate models making the simple assumption that MSA does not contribute to aerosol loading.

The study presents model sensitivity simulations exploring a range of mechanisms that might allow MSA to contribute to aerosol. In doing so, the authors highlight the sparcity of actual observations and lab measurements of MSA, which could be used to constrain the model simulations.

When allowing MSA to condense and nucleate in the model, only modest global ra-

diative effects (up to -40 mWm-2) are demonstrated. Regionally (e.g. in the Southern Ocean), larger differences in radiative effects are shown.

The study is subject to large uncertainties arising from the lack of measurements, as well as caveats introduced by the modelling approach. These uncertainties notwithstanding, the manuscript is a valid contribution to the field and should lead to further insights about the role of MSA.

Major comments

Please explain the rationale supporting the use multiple anthropogenic NH3 emissions inventories across different regions. It's not clear why this was done and what impact this may have on the NH3 concentrations and subsequently the MSA sensitivity tests.

The comparison method described in Section 2.5 assumes that the total sulphate will be additive between scenarios (i.e. substracting DEFAULT_NoMSA from the sensitivity simulations leaves the MSA-related contribution). This is an imperfect approach as the additional mass from MSA will grow the aerosol size distribution, and therefore increase rates of dry deposition and nucleation scavenging. This limitation should be noted in the method description. I think the comparison remains useful, however.

For consistency with the ATom measurements, was the model data used in the comparison also restricted to the sub-micron size range?

Page 11 line 18 refers to 'Table 4', which has not been provided. The data does appear in Figure 8.

Page 12 line 30 refers to Figure S7, but I think should refer to Figure S6. Please further check labelling of N10 / N80, and for left / right labelling in caption of Figure S6

Minor comments

There are typographic errors through the manuscript that a thorough read-through should reveal, including some sentences that have redundant words, or fragments from

previous iterations. I haven't attempted to highlight them.

Page 2 line 29: 'from marine particles' should be 'of marine particles'? Or 'from marine emissions'?

Page 3 line 1: VOCs – acronym not defined

Page 3 line 5/6: 'are an important source of marine emissions' should be 'are an important contributor to marine aerosol' (or something similar)? 'Aerosol' and 'particles' is being used in the same sense as 'emissions', which is incorrect.

Page 3 line 13: please state the relative yields of each product of DMS oxidation

Page 5 line 24: not clear what 'Regional EDGAR overwrites' are

Page 6 line 25: for completeness, please clarify whether MSA is assumed to be involved in the binary nucleation process

---

## Referee Comment (RC2) · Anonymous Referee #3 · 8 Feb 2019

This manuscript presents a sensitivity study estimating the potential influence of MSA, produced from oceanic DMS emission, on the submicron aerosol population and further on aerosol radiative effects in the global atmosphere. The paper relies on a set of global model simulations that cover the plausible range of parameters anticipated to affect how MSA contributes to the investigated issues. The used model has been evaluated previously in many other applications, so it can be considered appropriate for the purposes of this study. The paper is well organized, and the authors adequately discuss associated uncertainties. The conducted study itself is original and important to the scientific community.

[Figure]

I have a few, rather minor, issues to be considered before accepting this paper for publication.

The last sentence of page 2 (lines 28-31) is strange. Please modify.

Strictly speaking, primary biological or organic particles should not be called "organic compounds".

Are the latest estimates on the contribution of DMS to biogenic sulfur budget and sulfur precursor emission really as far back in time as from years 1990 and 2006. I also wonder the relative accuracy of the given numbers, i.e. 50% versus 21% (page 3).

The authors use rather old binary and ternary nucleation schemes in their simulations, together with a fixed tuning factor that may or may not be valid in marine environments that are more interesting than continental regions in this study. The authors investigated the sensitivity of their results on different assumptions on whether MSA participates on nucleation or not, but do not discuss whether these results are sensitive to apparent uncertainties in the nucleation scheme itself. I would like the authors to address this point at least by discussing it shortly.

The authors should discuss more explicitly what part of aerosol-cloud interactions they are attempting to capture in their simulations. Is it the first indirect effect only or something else as well?

---

## Author Response (AR1)

The manuscript by Hodshire et al uses a global aersol model to quantify the role of MSA in the aerosol-climate system. The role of MSA has not been quantified previously, with the majority of current aerosol and climate models making the simple assumption that MSA does not contribute to aerosol loading. The study presents model sensitivity simulations exploring a range of mechanisms that might allow MSA to contribute to aerosol. In doing so, the authors highlight the sparcity of actual observations and lab measurements of MSA, which could be used to constrain the model simulations. When allowing MSA to condense and nucleate in the model, only modest global radiative effects (up to -40 mWm-2) are demonstrated. Regionally (e.g. in the Southern Ocean), larger differences in radiative effects are shown. The study is subject to large uncertainties arising from the lack of measurements, as well as caveats introduced by the modelling approach. These uncertainties notwithstanding, the manuscript is a valid contribution to the field and should lead to further insights about the role of MSA.

**Major comments**

Please explain the rationale supporting the use multiple anthropogenic NH3 emissions inventories across different regions. It's not clear why this was done and what impact this may have on the NH3 concentrations and subsequently the MSA sensitivity tests.

We realized that this section is misleading as it is currently written. We have updated the text regarding inventories in general (below). We use ammonia overwrites as we expect these national and/or regional inventories to be more accurate than the older GEIA inventory.

**"Anthropogenic emissions except for ammonia, black carbon, and organic aerosol are from the Emissions Database for Global Atmospheric Research (EDGAR; Janssens-Maenhout et al., 2010). In Europe, Canada, the U.S., and Asia, anthropogenic emissions are overwritten by the European Monitoring and Evaluation Programme (Centre on Emissions Inventories and Projections, 2013), the Criteria Air Contaminant Inventory (http://www.ec.gc.ca/air/default.asp? lang=En&n=7C43740B-1), the National Emission Inventory from the U.S. EPA ((http://www.epa.gov/ttnchie1/net/2011inventory.html), and the MIX (Li et al., 2017) inventories, respectively. Black and organic carbon emissions from fossil-fuel and biofuel combustion processes are from Bond et al. (2007). Grid-box gas-phase concentrations of $NH_3$ are used in determining the volatility regime of MSA in the MSA parameterization (Sect. 2.2): global anthropogenic, biofuel, and natural ammonia**

**sources are from the Global Emissions InitiAtive (GEIA) (Bouwman et al., 1997). Anthropogenic ammonia emissions are overwritten over Europe, Canada, the U.S., and Asia using the same regional inventories discussed above for these regions. Ammonia emission from biomass burning are from FINNv1 (above)."**

The comparison method described in Section 2.5 assumes that the total sulphate will be additive between scenarios (i.e. substracting DEFAULT_NoMSA from the sensitivity simulations leaves the MSA-related contribution). This is an imperfect approach as the additional mass from MSA will grow the aerosol size distribution, and therefore increase rates of dry deposition and nucleation scavenging. This limitation should be noted in the method description. I think the comparison remains useful, however.

We agree with the reviewer that this is an important caveat to point out and have added the following to sect 2.5:

"We compare our sensitivity simulations to the ATom data as follows: we subtract the DEFAULT_NoMSA sulfate mass (that accounts for sulfate and sulfuric acid from DMS/SO$_2$ oxidation but not MSA) for the months of August (ATom-1) and February (ATom-2) from the sulfate mass for the months of August and February for each sensitivity case that includes MSA for each grid box. The resultant differences in sulfate mass represents the model-predicted contributions of MSA to the total sulfur budget for each case. **This is an imperfect approach, as the additional aerosol mass from the contribution of MSA will change the size distribution, therefore changing rates of wet and dry deposition, and is a limitation of this study."**

For consistency with the ATom measurements, was the model data used in the comparison also restricted to the sub-micron size range?

This is a reasonable point. We did not use the sub-micron range only; however, an off-line comparison shows that the percent difference between using the entire range (up to 10 μm) and the submicron range is well under 1%. We add the following to the text:

"We then compare the measured and predicted MSA mass by first averaging every ATom data point that falls within a given GC-TOMAS grid box. We then compare each averaged data point to that model grid box. The ATom data used in our analysis lies within 150-180° W (the Pacific ocean basin) and 10-40° W (the Atlantic ocean basin), and thus we use zonal averages of these longitude bands for both the ATom data and the GC-TOMAS output. We note that comparing monthly mean simulated values from 2014 to airborne measurements from a single point in time in 2016 and 2017 contributes to the apparent simulation errors. **We also note that we use the**

**full size range (3 nm -10 μm) of sulfate from the model output whereas the ATom data is submicron. However, the model-predicted percent difference in MSA mass between the full range and the submicron mass is well under 1% (not shown).**"

Page 11 line 18 refers to 'Table 4', which has not been provided. The data does appear in Figure 8.

Thank you for catching this--table 4 had been removed as it was redundant with the data in Figure 8. We have updated the text and made sure that no other references to Table 4 still exist.

Page 12 line 30 refers to Figure S7, but I think should refer to Figure S6. Please further check labelling of N10 / N80, and for left / right labelling in caption of Figure S6

Thank you for catching these errors, as well. The reference to Fig. S7 has been updated to Fig. S6. The label for Figure S6 has been updated to N80.

**Minor comments**

There are typographic errors through the manuscript that a thorough read-through should reveal, including some sentences that have redundant words, or fragments from C2 ACPD Interactive comment Printer-friendly version Discussion paper previous iterations. I haven't attempted to highlight them.

We have read throughout the text and corrected what typographical errors we have found.

Page 2 line 29: 'from marine particles' should be 'of marine particles'? Or 'from marine emissions'?

It should be 'of marine particles'--we have updated the text.

Page 3 line 1: VOCs – acronym not defined

Thank you--we have provided the full name along with the acronym.

Page 3 line 5/6: 'are an important source of marine emissions' should be 'are an important contributor to marine aerosol' (or something similar)? 'Aerosol' and 'particles' is being used in the same sense as 'emissions', which is incorrect.
We have updated the text:

**"Sulfur-containing organic compounds in the form of dimethylsulfide (DMS; CH$_3$SCH$_3$) and organosulfates (Bates et al., 1992, Quinn et al., 2015) are important precursors and contributors to marine aerosol."**

Page 3 line 13: please state the relative yields of each product of DMS oxidation

The relative yields of each product from DMS oxidation are uncertain. We do state in the methods that for this study, we assume that we follow the findings of Chin et al. (1996) and use a branching ratio of 75:25 for SO$_2$:MSA. However, this is a source of uncertainty. For instance, Chang et al. (2011) states: "The sensitivity of model results to uncertainties in DMS chemistry was also untested. In the oxidation of DMS by OH addition, the branching ratio between SO2 and methane sulfonic acid (MSA) is uncertain and, following Chin et al.[1996], we used a value of 75:25. Laboratory studies report percent yields of SO2:MSA of 65:4 [Yin et al.,1990a], 27:6 [Sørensen et al., 1996] and 38:11 [Arsene et al.,2001], for example."
We add the following to the introduction:

"The main products of DMS from oxidation by the hydroxyl radical are sulfur dioxide (SO$_2$) and methanesulfonic acid (CH$_3$S(O)$_2$OH, MSA) (Andreae et al., 1985). SO$_2$ can further oxidize to create sulfuric acid (H$_2$SO$_4$). **The relative yields of SO$_2$ and MSA from DMS oxidation are still uncertain, with reported branching ratios from oxidation of DMS by OH addition of SO$_2$:MSA varying across 75:25, 65:4, 27:6, and 38:11 (Yin et al., 1990; Chin et al., 1996; Sørensen et al., 1996; Arsene et al., 2001)."**

We also add text in the methods:

"In the standard GEOS-Chem DMS mechanism, DMS reacts with OH through the OH addition pathway to form molar yields of 0.75 SO$_2$ and 0.25 MSA (Chatfield and Crutzen, 1990; Chin et al., 1996). **As discussed in the introduction, laboratory studies have reported variable yields of SO$_2$ and MSA from DMS oxidation by OH addition. We do not test the sensitivity of our simulations to other pathways, and this is a source of uncertainty."**

Arsene, C., Barnes, I., Becker, K. H. and Mocanu, R.: FT‑IR product study on the photo‑oxidation of dimethyl sulphide in the presence of NOx—Temperature dependence, Atmos. Environ., 35(22), 3769–3780, doi:10.1016/S1352-2310(01)00168-6, 2001.

Chang, R. Y.‑W., Sjostedt, S. J. , Pierce, J. R., Papakyriakou, T. N., Scarratt, M. G., Michaud, S., Levasseur, M., Leaitch, W. R., and Abbatt, J. P. D.: Relating atmospheric and oceanic DMS levels to particle nucleation events in the Canadian Arctic, J. Geophys. Res., 116, D00S03, doi:10.1029/2011JD015926, 2011.

Chin, M., Jacob, D. J., Gardner, G. M., Foreman-fowler, M. S., Spiro, P. A. and Savoie, D. L.: A global three-dimensional model of tropospheric sulfate acid, , 101, doi:10.1029/ 96JD01221, 1996.

Sørensen, S., Falbe‑Hansen, H., Mangoni, M., Hjorth, J., and. Jensen, N. R: Observation of DMSO and CH3S(O)OH from the gas phase reaction between DMS and OH, J. Atmos. Chem., 24(3), 299–315, doi:10.1007/BF00210288, 1996.

Yin, F., D. Grosjean, R. C. Flagan, and J. H. Seinfeld: Photooxidation of dimethyl sulfide and dimethyl disulfide. II: Mechanism evaluation, J. Atmos. Chem., 11(4), 365–399, doi:10.1007/BF00053781, 1990.

Page 5 line 24: not clear what 'Regional EDGAR overwrites' are

We have updated this section in response to an early comment and refer the reviewer to that response.

Page 6 line 25: for completeness, please clarify whether MSA is assumed to be involved in the binary nucleation process

Thank you, we have added this clarification.

**When MSA is assumed to participate in nucleation, it is treated as an extra source of sulfuric acid for the ternary and binary nucleation schemes within the model.**

**Anonymous referee #3**

This manuscript presents a sensitivity study estimating the potential influence of MSA, produced from oceanic DMS emission, on the submicron aerosol population and further on aerosol radiative effects in the global atmosphere. The paper relies on a set of global model simulations that cover the plausible range of parameters anticipated to affect how MSA contributes to the investigated issues. The used model has been evaluated previously in many other applications, so it can be considered appropriate for the purposes of this study. The paper is well organized, and the authors adequately discuss associated uncertainties. The conducted study itself is original and important to the scientific community.

I have a few, rather minor, issues to be considered before accepting this paper for publication.

The last sentence of page 2 (lines 28-31) is strange. Please modify.

We agree and have rewritten this statement as follows:

 **To improve model estimates of the DRE and AIE, models must account for nucleation and condensational growth of marine particles.**

Strictly speaking, primary biological or organic particles should not be called "organic compounds".

We have updated the text:

**Biologically productive oceans emit volatile organic compounds (VOCs), primary biological particles, primary organic particles, and halocarbons (Quinn et al., 2015).**

Are the latest estimates on the contribution of DMS to biogenic sulfur budget and sulfur precursor emission really as far back in time as from years 1990 and 2006. I also wonder the relative accuracy of the given numbers, i.e. 50% versus 21% (page 3).

We appreciate the reviewer pointing out the outdatedness of these statistics. We have found more appropriate figures and have updated the text and references as follows:

**DMS accounts for approximately one-fifth of the global sulfur budget (Fiddes et al., 2017), with DMS flux estimates range from 9 to 35 Tg yr$^{-1}$ of sulfur (Belviso et al., 2004; Elliott, 2009; Woodhouse et al., 2010; Tesdal et al., 2016), although global DMS fluxes remain uncertain (Tesdal et al., 2016; Royer et al., 2015).**

Belviso, S., Bopp, L., Moulin, C., Orr, J. C., Anderson, T. R., Aumont, O., Chu, S., Elliott, S., Maltrud, M. E., and Simó, R.: Comparison of global climatological maps of sea surface dimethyl sulfide, Global Biogeochem. Cy., 18, GB3013, https://doi.org/10.1029/2003GB002193, 2004.

Elliott, S.: Dependence of DMS global sea-air flux distribution on transfer velocity and concentration field type, J. Geophys. Res.- Biogeo., 114, 1–18, https://doi.org/10.1029/2008JG000710, 2009.

Royer, S. J., Mahajan, A. S., Galí, M., Saltzman, E., and Simõ, R.: Small-scale variability patterns of DMS and phytoplankton in surface waters of the tropical and subtropical Atlantic, Indian, and Pacific Oceans, Geophys. Res. Lett., 42, 475–483, https://doi.org/10.1002/2014GL062543, 2015.

Sheng, J. X., Weisenstein, D. K., Luo, B. P., Rozanov, E., Stenke, A., Anet, J., Bingemer, H., and Peter, T.: Global atmospheric sulfur budget under volcanically quiescent conditions: Aerosol-chemistry-climate model predictions and validation, J. Geophys. Res.-Atmos., 120, 256–276, https://doi.org/10.1002/2014JD021985, 2015.

Tesdal, J. E., Christian, J. R., Monahan, A. H., and Von Salzen, K.: Evaluation of diverse approaches for estimating sea-surface DMS concentration and air-sea exchange at global scale, Environ. Chem., 13, 390–412, https://doi.org/10.1071/EN14255, 2016.

Woodhouse, M. T., Carslaw, K. S., Mann, G. W., Vallina, S. M., Vogt, M., Halloran, P. R., and Boucher, O.: Low sensitivity of cloud condensation nuclei to changes in the sea-air flux of dimethyl-sulphide, Atmos. Chem. Phys., 10, 7545–7559, https://doi.org/10.5194/acp-10-7545-2010, 2010.

The authors use rather old binary and ternary nucleation schemes in their simulations, together with a fixed tuning factor that may or may not be valid in marine environments that are more interesting than continental regions in this study. The authors investigated the sensitivity of their results on different assumptions on whether MSA participates on nucleation or not, but do not discuss whether these results are sensitive to apparent uncertainties in the nucleation scheme itself. I would like the authors to address this point at least by discussing it shortly.

We have added the following discussion to Sect. 2.6 (Study Caveats):
**We do not test the sensitivity of our simulations to the binary and ternary nucleation schemes used in this study, including potential sensitivity to the global tuning factor of $10^{-5}$ that was developed for continental regions (Jung et al., 2010; Westervelt et al., 2013). This source of uncertainty should be tested in future studies, as well.**

The authors should discuss more explicitly what part of aerosol-cloud interactions they are attempting to capture in their simulations. Is it the first indirect effect only or something else as well?

We apologize for the lack of clarity here: the aerosol-cloud interactions should be the cloud-albedo AIE, not just the AIE. We have amended the text to read 'cloud-albedo AIE' throughout, and have informed the reader that figure labels of 'AIE' refer to the cloud-albedo AIE, as well.

[revised manuscript text omitted]
 important precursors and contributors to marine aerosol. DMS accounts for approximately one-fifth of the global sulfur budget (Fiddes et al., 2017), with DMS flux estimates ranging from 9 to 35 Tg yr$^{-1}$ of sulfur (Belviso et al., 2004; Elliott, 2009; Woodhouse et al., 2010; Tesdal et al., 2016), although global DMS fluxes remain uncertain (Tesdal et al., 2016; Royer et al., 2015). DMS and its oxidation products have been the focus of many studies determining the gas-phase chemistry (e.g. Barnes et al. 2006 and references therein), gas-phase kinetics (e.g. Wilson and Hirst, 1996 and references therein), and possible impact to the aerosol size distribution and radiative budget (e.g. Korhonen et al., 2008; Woodhouse et al., 2013). Much of this research has stemmed from efforts to test the hypothesis that DMS emissions may regulate climate through a temperature-emissions feedback (the CLAW hypothesis, Charlson et al. (1987)).

The main products of DMS from oxidation by the hydroxyl radical are sulfur dioxide ($SO_2$) and methanesulfonic acid ($CH_3S(O)_2OH$, MSA) (Andreae et al., 1985). $SO_2$ can further oxidize to create sulfuric acid ($H_2SO_4$). The relative yields of $SO_2$ and MSA from DMS oxidation are still uncertain, with reported branching ratios from oxidation of DMS by OH addition of $SO_2$:MSA varying across 75:25, 65:4, 27:6, and 38:11 (Yin et al., 1990; Chin et al., 1996; Sørensen et al., 1996; Arsene et al., 2001). 
[revised manuscript text omitted]

Anna Lily Hodshire 2/12/2019 9:25 PM

Anna Lily Hodshire 2/12/2019 9:25 PM

inventories discussed above for these regions. Ammonia emission from biomass burning are from FINNv1 (above). All simulations are run for 2014, with one month of model spinup that is not included in the analysis. All results are presented as annual or monthly averages.

We use the default (at the time of this model version) GEOS-Chem DMS emissions inventory (Kettle et al., 1999; Kettle and Andreae 2000) for this study. We acknowledge that the updated DMS inventory of Lana et al. (2011) includes more up-to-date measurements than the default DMS inventory for GEOS-Chem v10.01. Their work found that the default climatology overpredicted DMS emissions in some latitudes/seasons but underpredicted DMS emissions in other latitudes/seasons. We found, however, that using the Lana emission inventory led to minor differences in MSA impacts spatially but overall, similar magnitudes of changes were observed. The supplement Sect. S2 provides more analysis of the two different emissions inventories.

In the standard GEOS-Chem DMS mechanism, DMS reacts with OH through the OH addition pathway to form molar yields of 0.75 $SO_2$ and 0.25 MSA (Chatfield and Crutzen, 1990; Chin et al., 1996). As discussed in the introduction, laboratory studies have reported variable yields of $SO_2$ and MSA from DMS oxidation by OH addition. We do not test the sensitivity of our simulations to other pathways, and this is a source of uncertainty. 
[revised manuscript text omitted]
. MSA data from the third and fourth ATom missions, ATom-3 and ATom-4, were not used in this study, but the calibration details discussed in Sect. S5 apply to these missions, as well. Overall sensitivity (as determined daily from the ionization efficiency of nitrate, $IE_{NO3}$), relative ionization efficiencies and particle transmission (all determined periodically in the field) were stable over all four deployments. Particle phase MSA concentrations for all ATom flights are reported based on the intensity of the highly specific marker ion $CH_3SO_2^+$ (Phinney et al, 2006, Zorn et al, 2008). The quantification of MSA $PM_1$ concentrations from the signal intensity of the $CH_3SO_2^+$ fragment is described in detail in the SI, Sect. S5. Positive Matrix Factorization (Paatero 1994; Ulbrich et al., 2009) of the ATom-1 organic aerosol (OA) and sulfate data confirmed the specificity of the marker ion for MSA and the consistency of the field mass spectra with those acquired in the MSA calibrations. Importantly, it also confirmed that the AMS response to MSA is independent of the aerosol acidity, which varied significantly over the range of conditions found in ATom. Further details are provided in Sect S5.

For the data presented here, the AMS raw data was processed at 1 minute resolution. Under those conditions, the detection limit of MSA was in the range 1.5-3 ng sm$^{-3}$ (0.3-0.6 pptv), and will decrease with the square root of the number of

Anna Hodshire 2/13/2019 2:59 PM

Anna Hodshire 2/13/2019 2:59 PM

Anna Hodshire 2/13/2019 2:59 PM

Anna Hodshire 2/13/2019 2:59 PM

Anna Lily Hodshire 2/13/2019 9:10 PM

Anna Lily Hodshire 2/13/2019 9:12 PM

averaged 1-minute data points. The uncertainty in the MSA quantification as detailed in the SI, Sect. S5, is comparable to that of sulfate, hence the overall uncertainty in the quantification is estimated to be +/-35% (2 standard deviations; Bahreini et al., 2009).

We compare our sensitivity simulations to the ATom-1 and ATom-2 data as follows: we subtract the DEFAULT_NoMSA sulfate mass (that accounts for sulfate and sulfuric acid from DMS/$SO_2$ oxidation but not MSA) for the months of August (ATom-1) and February (ATom-2) from the sulfate mass for the months of August and February for each sensitivity case that includes MSA for each grid box. The resultant differences in sulfate mass represents the model-predicted contributions of MSA to the total sulfur budget for each case. This is an imperfect approach, as the additional aerosol mass from the contribution of MSA will change the size distribution, therefore changing rates of wet and dry deposition, and is a limitation of this study. We then compare the measured and predicted MSA mass by first averaging every ATom data point that falls within a given GC-TOMAS grid box. We then compare each averaged data point to that model grid box. The ATom data used in our analysis lies within 150-180° W (the Pacific ocean basin) and 10-40° W (the Atlantic ocean basin), and thus we use zonal averages of these longitude bands for both the ATom data and the GC-TOMAS output. We note that comparing monthly mean simulated values from 2014 to airborne measurements from a single point in time in 2016 and 2017 contributes to the apparent simulation errors. We also note that we use the full size range (3 nm -10 μm) of sulfate from the model output whereas the ATom data is submicron. However, the model-predicted percent difference in MSA mass between the full range and the submicron mass is well under 1% (not shown).

To evaluate model performance, we calculate the log-mean bias (LMB), the slope of the log-log regression ($m$), and the coefficient of determination ($R^2$) between each cosampled GC-TOMAS grid box and averaged measurement point that falls within that GC-TOMAS grid box. The LMB is calculated through:

$$LMB = \frac{\sum_i^N (log_{10}(S_i) - log_{10}(O_i))}{N},\qquad(3)$$

where $S_i$ and $O_i$ are the simulated and observed MSA masses, respectively, for each data point $i$, and $N$ is the number of data points. A LMB of 1 means that on average, the model overestimates the measurements by a factor of $10^1$ (10); a LMB of -1 means that on average, the model underestimates the measurements by a factor of $10^{-1}$ (0.1); a LMB of 0 indicates no bias between the model and measurements ($10^0$ = 1.00). LMB, $m$, and $R^2$ are summarized in Fig. 8 (discussed in Sect. 3.4). Since MSA is observed only in the particle-phase in the ATom measurements, we do not include the NoDMS_NoMSA (no DMS emissions in the model) sensitivity case in our analysis of the ATom data. We present the aggregated results of the two campaigns, as well as results for each campaign and ocean basin. The ATom-1 mission provided more data points than the ATom-2 missions (1258 vs. 1000) and thus the aggregate results are slightly skewed towards the ATom-1 results.

**2.6 Study caveats**

This study is intended to examine the sensitivity of the aerosol size distribution and radiative impacts implied by the various sensitivity treatments of MSA (Table 2). However, our treatments of DMS and MSA still fall short of what is

Anna Hodshire 2/13/2019 3:03 PM

Anna Lily Hodshire 2/12/2019 10:16 PM

currently known about organic condensational behavior. Assuming idealized semivolatile condensation with no re-evaporation due to conditional changes (e.g. change in temperatures, RH) may overestimate the amount of MSA able to condense on particles; but it may also underestimate particle-phase MSA if conditions for condensation switch from unfavorable to favorable after MSA chemical production. Further, relying on E-AIM simulations to construct our volatility parameterization could have hidden biases due to an incomplete understanding of the system. We are also neglecting known as well as gas-phase and aqueous-phase oxidation pathways of DMS that are currently not included in GEOS-Chem. The standard GEOS-Chem model does not include DMS oxidation through the OH or halogen addition pathways to dimethylsulfoxide (DMSO). DMSO chemistry reduces the yield of sulfate formation from $DMS/SO_2$ oxidation (Breider et al., 2014) by increasing the yields of both gas-phase and aqueous phase MSA as well as aqueous-phase dimethyl sulfone ($DMSO_2$), another stable oxidation product (Hoffmann et al. 2016). To reduce the number of parameters for this study, we do not include the DMSO pathway. We acknowledge that neglecting this pathway will slightly bias our estimates of the contributions to the aerosol size distribution of sulfate and MSA mass high and low, respectively. Further, aqueous-phase production of MSA would condense on CCN-sized particles, similar to aqueous phase sulfate (Sect 2.1), shifting the size distribution to larger sizes. Heterogeneous oxidation may limit the lifetime of MSA in the particle phase (Mungall et al., 2017; Kwong et al., 2018), although the reactive uptake coefficients from these studies are somewhat dissimilar, indicating a need for further study of the system. Regardless, neglecting heterogeneous chemistry could overestimate the estimate of the contribution of MSA to aerosol mass. Finally, if MSA does participate in nucleation, it is unlikely that it will behave exactly like sulfuric acid, as it is treated here. All of the limitations described above are important and require further testing in detailed chemical models and chemical-transport models in order to determine their effects.

Another limitation of this study is our reliance upon the current ammonia inventory in GEOS-Chem as well as our cutoff value of 10 ppt of ammonia between the no ammonia and excess ammonia regimes (Sect. 2.2). Uncertainties in the ammonia inventories over the oceans could change our results, as could a different cutoff value. As this study is focused on MSA sensitivities, we will leave sensitivities of MSA to ammonia for a future study. It is important to note that other bases such as amines could also have an important effect on MSA's effective volatility (e.g. Chen and Finlayson-Pitts, 2017). However, the standard GEOS-Chem currently does not account for gas-phase bases beyond ammonia, and this sensitivity will also be left for a future study.

We do not test the sensitivity of our simulations to the binary and ternary nucleation schemes used in this study, including potential sensitivity to the global tuning factor of $10^{-5}$ that was developed for continental regions (Jung et al., 2010; Westervelt et al., 2013). This source of uncertainty should be tested in future studies, as well.

**3 Results and Discussion**

Figure 2 shows the global annual mean percent change (at 900 hPa and zonally) for submicron mass by adding MSA for the PARAM_NoNuc, ELVOC_NoNuc, SVOC_NoNuc, PARAM_Nuc, and ELVOC_Nuc simulations. Figure 3 shows the global annual mean percent change in N3 and N80 due to addition of MSA at 900 hPa and zonally for all model

Anna Lily Hodshire 2/13/2019 9:18 PM

Anna Lily Hodshire 2/13/2019 9:19 PM

Anna Lily Hodshire 2/12/2019 11:27 PM

levels for each of these cases, and Fig. 4 shows the corresponding global annual cloud-albedo AIE and DRE of MSA. Figure 5 shows the global annual mean percent contribution from DMS/$SO_2$ oxidation (at 900 hPa and zonally) alone (not including MSA) to submicron mass, N3, N80, AID, and DRE. Figure 6 and Table S3 summarises the results of Figs. 2, 3, 4, and 5. All of the numerical statistics presented in Sects. 3.1-3.4 are for the annual mean, either globally or between 30°-90°S. Each case with MSA is analyzed for the change relative to DEFAULT_NoMSA to determine the impact that MSA has on the size distribution and resulting radiative effects (positive values indicate that the inclusion of MSA increases a given metric). For reference, Figure S6 provides the absolute number concentration for N3 and N80 at 900 hPa and zonally for all model levels for the DEFAULT_NoMSA simulation. We will refer back to these figures in the following sections.

**3.1 Volatility-dependent impact of MSA if MSA does not participate in nucleation**

The top rows of Figs. 2 and 3 show the global annual mean percent change at 900 hPa and zonally from adding MSA using the volatility parameterization without nucleation (PARAM_NoNuc - DEFAULT_NoMSA) for submicron aerosol mass (Fig. 2) and N3 and N80 (Fig. 3). By adding MSA with these assumptions, we predict at 900 hPa an increase in submicron mass of 0.7% globally and 1.3% between 30°S-90°S; a decrease in N3 of -3.9% globally and -8.5% between 30°S-90°S; and an increase in N80 of 0.8% globally and 1.7% between 30°S-90°S (Fig. 6 and Table S3). These MSA impacts are limited by ammonia availability. Figures S1 and S2 show that many oceanic regions are predicted to have annual and seasonal ammonia mixing ratios of less than 10 ppt. Below 10 pptv of ammonia, MSA condensation as SVOC-like or VOC-like (no condensation) (Fig. 1a) and MSA condensation will only be SVOC-like if the RH > 90%; under these conditions for the majority of the year, MSA will be a VOC-like species over Antarctica (low RH conditions) and often an SVOC-like species over the southern-ocean boundary layer (high RH conditions). Only in the Southern Hemisphere (SH) winter months does ammonia exceed 10 ppt over appreciable regions in the southern oceans (Fig. S2); during this time, MSA condensation is ELVOC-like due to cold temperatures (Fig. 1b). As shown in D'Andrea et al. (2013), ideal-SVOC material largely condenses primarily to accumulation-mode particles, which in turn suppresses N3 through increased coagulation and reduced nucleation and has little impact on N80. In the midlatitudes, the annual and seasonal ammonia concentrations often exceed 10 ppt, and thus MSA condensation will be either ELVOC-like under low-temperature and/or high-RH conditions or SVOC-like under high-temperature and/or low-RH conditions. D'Andrea et al. (2013) showed that adding ELVOC material can increase N80 by increasing growth of ultrafine particles but also can suppress N3 through the same coagulation/nucleation feedbacks. This combination of ammonia-rich and ammonia-poor regions lead to MSA giving an overall weak increase in N80 with a large suppression of N3 in some regions. We note that these results are somewhat sensitive to the simulated ammonia concentrations and may be sensitivity to the ammonia cutoff of 10 ppt in the MSA-volatility parameterization. As there are already uncertainties in many other dimensions, we do not attempt to quantify the sensitivity of MSA towards ammonia in this work.

The idealized volatility cases, ELVOC_NoNuc (Figs. 2 and 3, second row) and SVOC_NoNuc (Figs. 2 and 3, third row) help to highlight and further explain MSA's volatility-dependent contribution towards growth. In both of these cases,

Jeffrey Pierce 2/13/2019 8:41 PM
Comment [1]: leave this for the caption
Jeffrey Pierce 2/13/2019 8:41 PM

Anna Lily Hodshire 2/12/2019 10:22 PM

[revised manuscript text omitted]

Jeffrey Pierce 2/13/2019 8:43 PM

Data for the ATom campaigns is posted publicly at https://doi.org/10.3334/ORNLDAAC/1581. The GEOS-Chem model is available at http://wiki.seas.harvard.edu/geos-chem/.

**Author contributions**

ALH, JRP, and BC defined the scientific questions and scope of this work. ALH and BC performed all GEOS-Chem model simulations and off-line calculations with help from JKK, BC, and JRP. JRP performed the E-AIM calculations. PCJ, BAN, JCS, and JLJ carried out the primary measurements and data processing for the ATom field campaign, as well as campaign supervision and design. ALH prepared the primary text with substantial contributions from JRP, JKK, BC, PCJ, and JLJ. PCJ provided the detailed description provided in the supplement of the calibration method used for detecting MSA during the ATom field campaign, with additional contributions from JLJ.

**Competing interests**

The authors declare that they have no competing interests.

**Acknowledgements**

This research was supported by the US Department of Energy's Atmospheric System Research, an Office of Science, Office of Biological and Environmental Research program, under Grant No. DE-SC0011780 and by the U.S National Oceanic and Atmospheric Administration, an Office of Science, Office of Atmospheric Chemistry, Carbon Cycle, and Climate Program, under the cooperative agreement awards #NA17OAR430001. B.C. was supported under the Climate Change and Atmospheric Research programme at 1164 NSERC, as part of the NETCARE project. The CU-Boulder group was supported by NASA NNX15AH33A and NNX15AJ23G.

[revised manuscript text omitted]